# Interfacial engineering of $Bi_2S_3/Ti_3C_2T_x$ MXene based on work function for rapid photo-excited bacteria-killing

Jianfang Li[1], Zhaoyang Li[1], Xiangmei Liu[2✉], Changyi Li[3], Yufeng Zheng[4], Kelvin Wai Kwok Yeung[5], Zhenduo Cui[1], Yanqin Liang[1], Shengli Zhu[1], Wenbin Hu[1], Yajun Qi[2], Tianjin Zhang[2], Xianbao Wang[2] & Shuilin Wu[1✉]

In view of increasing drug resistance, ecofriendly photoelectrical materials are promising alternatives to antibiotics. Here we design an interfacial Schottky junction of $Bi_2S_3/Ti_3C_2T_x$ resulting from the contact potential difference between $Ti_3C_2T_x$ and $Bi_2S_3$. The different work functions induce the formation of a local electrophilic/nucleophilic region. The self-driven charge transfer across the interface increases the local electron density on $Ti_3C_2T_x$. The formed Schottky barrier inhibits the backflow of electrons and boosts the charge transfer and separation. The photocatalytic activity of $Bi_2S_3/Ti_3C_2T_x$ intensively improved the amount of reactive oxygen species under 808 nm near-infrared radiation. They kill 99.86% of *Staphylococcus aureus* and 99.92% of *Escherichia coli* with the assistance of hyperthermia within 10 min. We propose the theory of interfacial engineering based on work function and accordingly design the ecofriendly photoresponsive Schottky junction using two kinds of components with different work functions to effectively eradicate bacterial infection.

[1] School of Materials Science & Engineering, The Key Laboratory of Advanced Ceramics and Machining Technology by the Ministry of Education of China, Tianjin University, Tianjin, China. [2] Hubei Key Laboratory of Polymer Materials, Ministry-of-Education Key Laboratory for the Green Preparation and Application of Functional Materials, School of Materials Science & Engineering, Hubei University, Wuhan, China. [3] Stomatological Hospital, Tianjin Medical University, Tianjin, China. [4] College of Engineering, State Key Laboratory for Turbulence and Complex System, Department of Materials Science and Engineering, Peking University, Beijing, China. [5] Department of Orthopaedics & Traumatology, Li Ka Shing Faculty of Medicine, The University of Hong Kong, Pokfulam Hong Kong, China. ✉email: liuxiangmei1978@163.com; shuilinwu@tju.edu.cn

Globally, infectious diseases impose an enormous burden on public health, the economy, and mental well-being. For instance, the worldwide outbreak of COVID-19 is seriously threatening global public health[1]. The latest clinical reports disclose that many COVID-19 patients die of secondary infections, including antibiotic-resistant bacterial infections, rather than the virus itself[2]. Once bacteria come in contact with a wound, they use the nutrients surrounding the wound and start growing at an alarming rate, infecting and inflaming the wound. This is especially true for diabetics[3]. Currently, antibiotics are used to treat bacterial infectious diseases. Unfortunately, systemic antibiotic therapy is often long-drawn and requires large doses of antibiotics, which not only impairs the immune system[4], but also leads to bacterial resistance and even the emergence of super bacteria[5]. Hence, exploring efficient and safe solutions without antibiotics is of great significance for treating bacteria-induced diseases.

Recently, photoresponsive biomedical materials have been shown to be promising alternatives for antibiotic-free therapy of bacterial infections, because hyperthermia or the radical oxygen species (ROS) produced by them under light can kill bacteria to some extent[6,7]. However, practically, the efficacies of ROS and heat generation decrease due to poor electron mobility and rapid recombination of electrons and holes in single-component semiconductors[8,9].

$Ti_3C_2T_x$ MXene is an ecofriendly material with an excellent advantage due to its photo-induced charge carriers[10–12]. The surface functional terminations of $Ti_3C_2T_x$ MXene can provide more active sites for semiconductors. It is an attractive photoresponsive material, given its localized surface plasmon resonance (LSPR), strong absorption and conversion efficiencies for near-infrared (NIR) light[13], and high electrical conductivity, which can be utilized to regulate the energy bands of semiconductors, thereby enhancing the photocatalytic properties by accelerating the photo-induced charge transfer[14,15]. Although $Ti_3C_2T_x$ MXene has exhibited excellent antibacterial performance at least 4 h and good biocompatibility at low concentration in the dark under continuous shaking condition[16–22], few studies reported the antibacterial activity under 808 nm NIR light.

Recently, many photocatalysts are used for photodynamic therapy, such as $C_3N_4$, black phosphorus, $TiO_2$, ZnO, $MoS_2$, $Bi_2S_3$, and so on. Compared with other semiconductors, $Bi_2S_3$ is considered as a potential photocatalytic material due to its n-type nature, direct narrow gap, and high absorption coefficient[23,24]. The excellent biocompatibility also broadens its application potential. However, the narrow band gap easily gives rise to rapid recombination of the photogenerated electrons and holes, reducing the utilization of the former[25–27]. Moreover, the intrinsic poor conduction usually cannot ensure effective photogenerated electron transport[28,29]. These deficiencies seriously mar its overall attractiveness despite the advantages conferred by its photocatalytic properties.

In view of the intrinsic physical characteristics of $Ti_3C_2T_x$ and $Bi_2S_3$ due to the different work functions (WF) values, we hypothesize that an interfacial self-driven charge transfer channel can be built by combining $Ti_3C_2T_x$ with $Bi_2S_3$ to form a Schottky junction to accelerate charge transmission at the interface. Based on this hypothesis, herein, $Bi_2S_3$ nanorods are grown in situ on the surface of $Ti_3C_2T_x$ nanosheets in this work. The fabrication process of the materials is schematically illustrated in Supplementary Fig. 1. Our results disclose that the combination of $Ti_3C_2T_x$ and n-type $Bi_2S_3$ boost the yield of ROS due to the accelerated photogenerated charge separation and transfer resulting from the contact potential difference between the two components with different WF values.

## Results

### Characterization of $Ti_3C_2T_x$, $Bi_2S_3$, and $Bi_2S_3$/$Ti_3C_2T_x$ Schottky catalyst

$Bi_2S_3$/$Ti_2C_3T_x$ was prepared with a two-step approach, which is schematically shown in Supplementary Fig. 1. The related morphology, microstructure, chemical composition, zeta potential, phase, size, and binding energy of the synthesized component were shown in Supplementary Figs. 2–10, respectively, and explained in detail below the corresponding figures. The transmission electron microscopy (TEM) results of $Bi_2S_3$/$Ti_3C_2T_x$ (Fig. 1a) showed that the clubbed $Bi_2S_3$ grains were closely anchored onto the nanosheets to ensure intimate contact. The corresponding TEM element mapping analysis indicated that the C, Ti, O, F, Bi, and S elements were homogeneously distributed across the hybrid (Fig. 1b). The high-resolution TEM (HRTEM) image (Fig. 1c) showed lattice fringes of 0.36 and 0.26 nm, corresponding to the d-spacing of the (130) crystal plane of the orthorhombic $Bi_2S_3$[30] and the $(0\bar{1}10)$ crystal plane of $Ti_3C_2T_x$[31], indicating the in situ formation of $Bi_2S_3$. This intimate mutual contact also effectively prevented the stacking or aggregation of individual $Ti_3C_2T_x$ nanosheets. The density function theory (DFT) calculation confirmed the existence of many different functional groups on the $Ti_3C_2T_x$ surface (Fig. 1d) and their strong interactions with cations, which facilitated the formation of the $Bi_2S_3$/$Ti_3C_2T_x$ complex. A stable crystal structure was formed and the average interface space was about 2.34 Å, confirming close contact[32,33].

### Photocatalytic and photothermal performance

Pure $Ti_3C_2T_x$ nanosheets produced only negligible ROS under 808 nm for 10 min (Fig. 2a). Conversely, the ROS yields produced from $Bi_2S_3$ and $Bi_2S_3$/$Ti_3C_2T_x$ gradually increased as the irradiation time increased, with $Bi_2S_3$/$Ti_3C_2T_x$-5 reaching the highest value. Thus, $Bi_2S_3$/$Ti_3C_2T_x$-5 exhibited the best NIR photocatalytic activity. The photocurrent densities of the different samples are shown in Fig. 2b. Compared to $Ti_3C_2T_x$, all of the $Bi_2S_3$/$Ti_3C_2T_x$ samples exhibited improved photocurrent densities, indicating enhanced spatial charge separation abilities[34]. $Bi_2S_3$/$Ti_3C_2T_x$-5 presented the largest photocurrent density, illustrating the best transfer and separation of photo-induced electrons at this ratio between $Ti_3C_2T_x$ and $Bi_2S_3$. The photoluminescence (PL) spectra (Fig. 2c) further certified this result. Obviously, a peak at 445 nm with the highest PL band was observed for $Bi_2S_3$. $Bi_2S_3$/$Ti_3C_2T_x$-5 showed the lowest PL intensity, indicating the lowest radiative recombination rate of photogenerated electron–hole pairs[35]. The impedance measured by electrochemical impedance spectroscopy (EIS) can be seen in Fig. 2d. The NIR groups showed smaller semicircles (indicating lower impedance) than these of non-NIR groups. $Bi_2S_3$/$Ti_3C_2T_x$-5 exhibited the smallest impedance among all the samples (the corresponding equivalent circuit and measured parameters are shown in Supplementary Fig. 11 and Supplementary Table 1, respectively). These results suggested that the smallest charge transfer resistance and fastest charge transmission occurred at the interface.

Locally elevated temperature can affect bacteria by changing the activity of their proteins, nucleic acids, cell walls, and cell membranes, further impacting metabolism[36]. The combination of $Ti_3C_2T_x$ with $Bi_2S_3$ enhanced the temperature after 10 min (Fig. 2e). $Bi_2S_3$/$Ti_3C_2T_x$-5 exhibited the best photothermal performance with the highest NIR thermal conversion efficacy under the optimal concentration of 200 ppm (Supplementary Figs. 12 and 13). The repeatable heating/cooling curves suggested that the prepared junction possessed excellent photothermal stability without obvious temperature change even after four irradiation cycles (Fig. 2f). The corresponding photothermal conversion efficiency ($\eta$) of $Bi_2S_3$/$Ti_3C_2T_x$-5 reached 35.43%

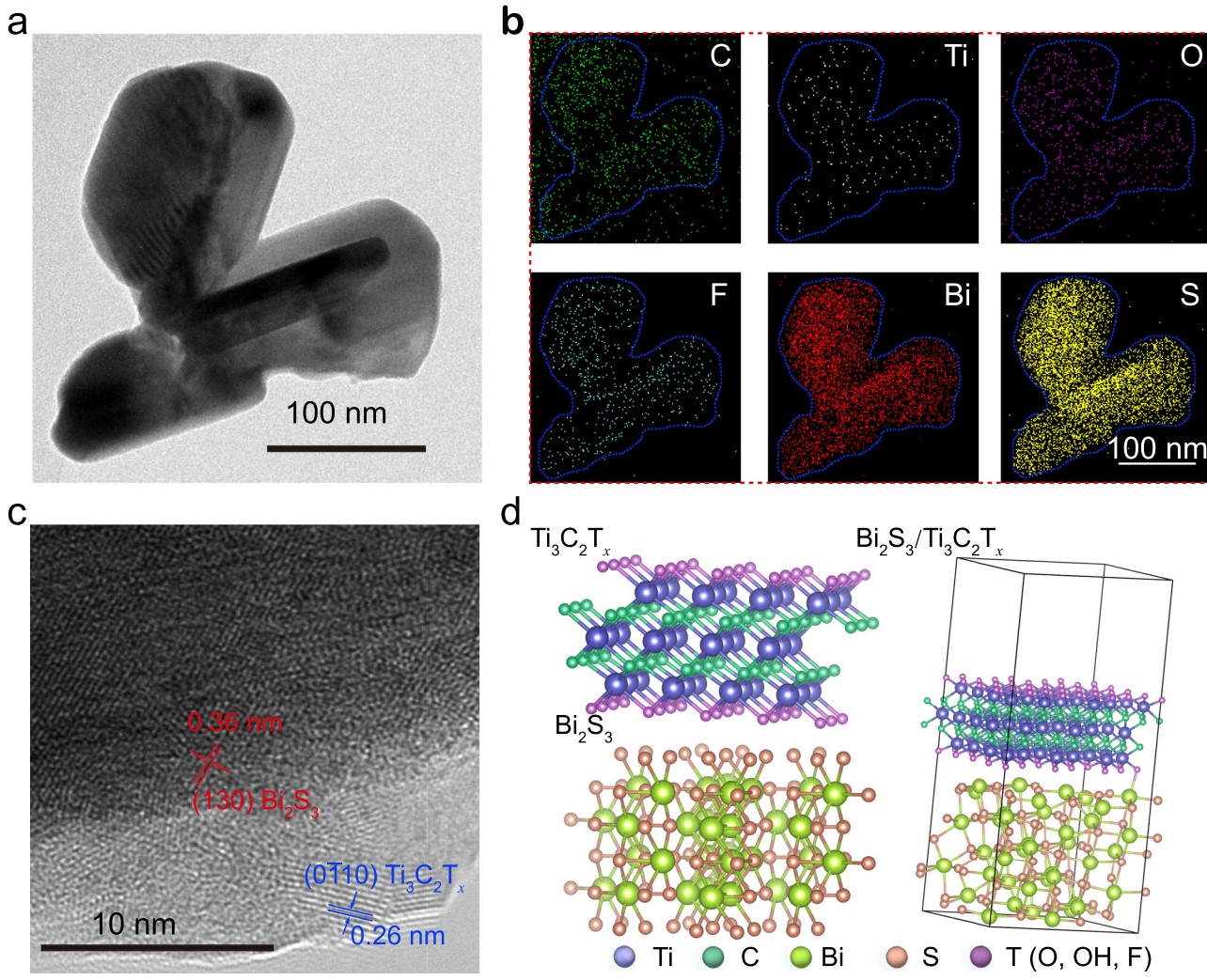

**Fig. 1 The characterizations of Ti$_3$C$_2$T$_x$/Bi$_2$S$_3$. a, b,** TEM image of Ti$_3$C$_2$T$_x$/Bi$_2$S$_3$ and EDS elemental mappings. **c,** HRTEM of Ti$_3$C$_2$T$_x$/Bi$_2$S$_3$. **d,** Crystal structure of Ti$_3$C$_2$T$_x$, Bi$_2$S$_3$ and Ti$_3$C$_2$T$_x$/Bi$_2$S$_3$ after structure optimization.

(Supplementary Fig. 14), and the calculated time constant was found to be 3.74 min by fitting the linear regression curve in the cooling period. This high photothermal conversion efficiency also unveiled the effectiveness of converting into thermal energy.

**Mechanism of photocatalytic performance.** According to the band theory, electron transfer behavior is closely related to the WF of Ti$_3$C$_2$T$_x$ and Bi$_2$S$_3$. Ultraviolet photoelectron spectroscopy (UPS) spectra were utilized to determine the work functions of the samples. The measured secondary electron cutoff energy ($E_{cutoff}$) of Ti$_3$C$_2$T$_x$ was 16.68 eV (Fig. 3a). WF was calculated to be 4.54 eV by subtracting the excitation energy of He I (21.22 eV). Similarly, an $E_{cutoff}$ value of 16.90 eV was used to deduce WF (4.32 eV) for Bi$_2$S$_3$. The energy maximum ($E_v$) of the valence band with respect to the Fermi level ($E_F$) of Bi$_2$S$_3$ was 1.25 eV. After establishing contact between Ti$_3$C$_2$T$_x$ and Bi$_2$S$_3$, $E_{cutoff}$ of Bi$_2$S$_3$/Ti$_3$C$_2$T$_x$-5 was 16.72 eV and the corresponding WF was 4.5 eV. In addition, the band gap between $E_v$ and the $E_F$ narrowed, ranging from 1.25 to 1.13 eV (Fig. 3a), indicating the increased concentration of holes[37]. Thus, the $E_v$ levels of Bi$_2$S$_3$ and Bi$_2$S$_3$/Ti$_3$C$_2$T$_x$-5 were calculated to be −5.57 and −5.63 eV, respectively. Under 808 nm NIR light irradiation, the electron spin resonance (ESR) spectra, Nitro Blue Tetrazolium (NBT) method, and 1,3-Diphenylisobenzofuran (DPBF) (Fig. 3b, c and Supplementary Fig. 15) were used to measure ·OH, ·O$_2^-$, $^1$O$_2$,

respectively. As it can be seen in Fig. 3b, c, obviously, ·OH and ·O$_2^-$ were detected under 808 nm NIR irradiation, while no signals for these two species were detected in the dark, indicating that ·OH and ·O$_2^-$ can be generated under 808 nm NIR light[38,39]. In addition, NBT can react with ·O$_2^-$ to produce monoformazan (MF) that exhibits a maximum absorbance at 530 nm. As shown in Supplementary Fig. 15a, compared with the dark group, Bi$_2$S$_3$/Ti$_3$C$_2$T$_x$-5 under 808 nm NIR light irradiation had high absorption, suggesting the production of ·O$_2^-$. DPBF result indicated that almost nothing $^1$O$_2$ was detected during 10 min NIR light irradiation (Supplementary Fig. 15b).

Based on the above wet-lab results, we can make the following conclusions. Before contact between Ti$_3$C$_2$T$_x$ and Bi$_2$S$_3$, according to the result of band gap (1.56 eV; deduced from the ultraviolet-visible-NIR diffuse reflectance spectra shown in Supplementary Fig. 16), the $E_F$ of Bi$_2$S$_3$ was calculated to be close to its conduction band, indicating the formation of an *n*-type semiconductor. The $E_F$ of Bi$_2$S$_3$ was higher than that of Ti$_3$C$_2$T$_x$. These data suggest a Schottky junction can be formed at the interface between Bi$_2$S$_3$ and Ti$_2$C$_3$T$_x$. After contact between Ti$_3$C$_2$T$_x$ and Bi$_2$S$_3$, the electrons spontaneously flowed from Bi$_2$S$_3$ to Ti$_3$C$_2$T$_x$ because of the higher $E_F$ of Bi$_2$S$_3$ until a balance was reached. The result also can be confirmed by the peak shifts of the XPS. During this process, a positively charged layer on the Bi$_2$S$_3$ surface was created due to the missing electrons, resulting in the

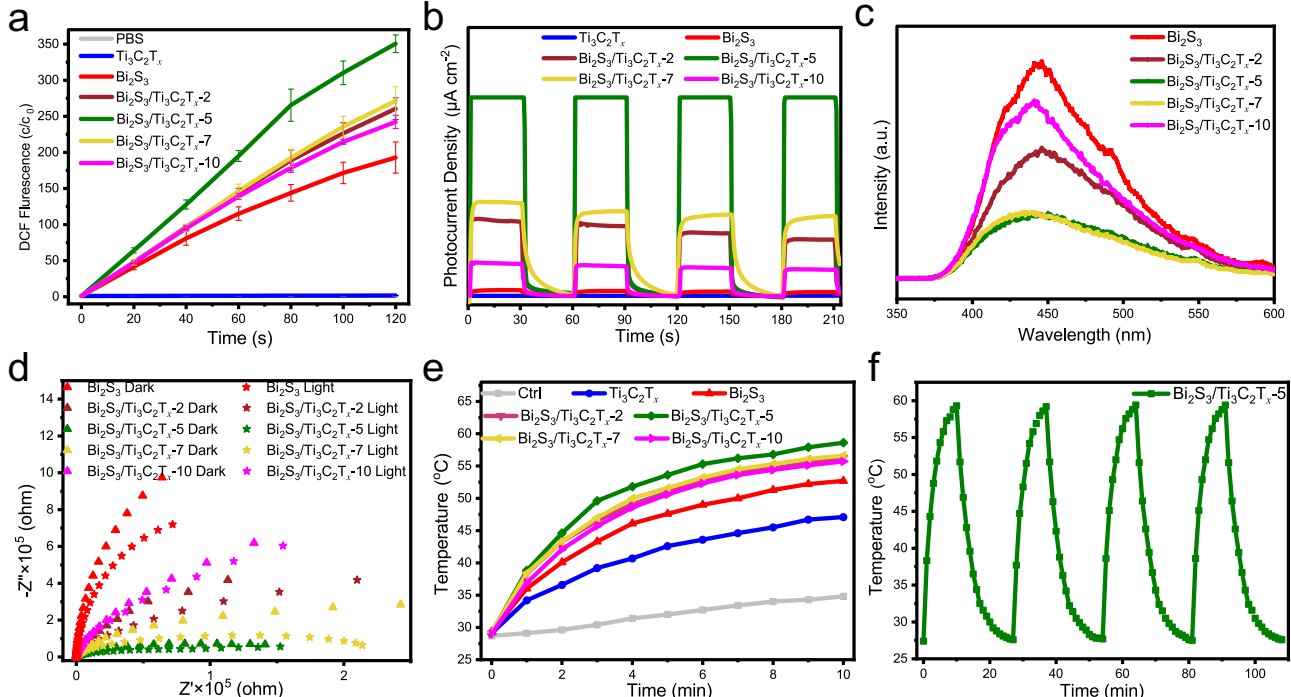

**Fig. 2 The photocatalytic and photothermal properties of Ti$_3$C$_2$T$_x$, Bi$_2$S$_3$, Bi$_2$S$_3$/Ti$_3$C$_2$T$_x$-2, Bi$_2$S$_3$/Ti$_3$C$_2$T$_x$-5, Bi$_2$S$_3$/Ti$_3$C$_2$T$_x$-7, and Bi$_2$S$_3$/Ti$_3$C$_2$T$_x$-10.** **a** ROS production with DCFH fluorescence probe (200 ppm). Data are presented as mean ± standard deviations from a representative experiment ($n = 3$ independent samples). **b** Photocurrent response under 808 nm NIR irradiation. **c** PL spectra tested with 325 nm excitation wavelength from 350 nm to 600 nm. **d** EIS tests with or without 808 nm irradiation. **e** Photothermal curves of different samples under 10 min light (808 nm) irradiation. **f** Temperature rising and cooling profiles of Bi$_2$S$_3$/Ti$_3$C$_2$T$_x$-5 when the irradiation is on/off under 808 nm. Source data are provided as a Source Data file.

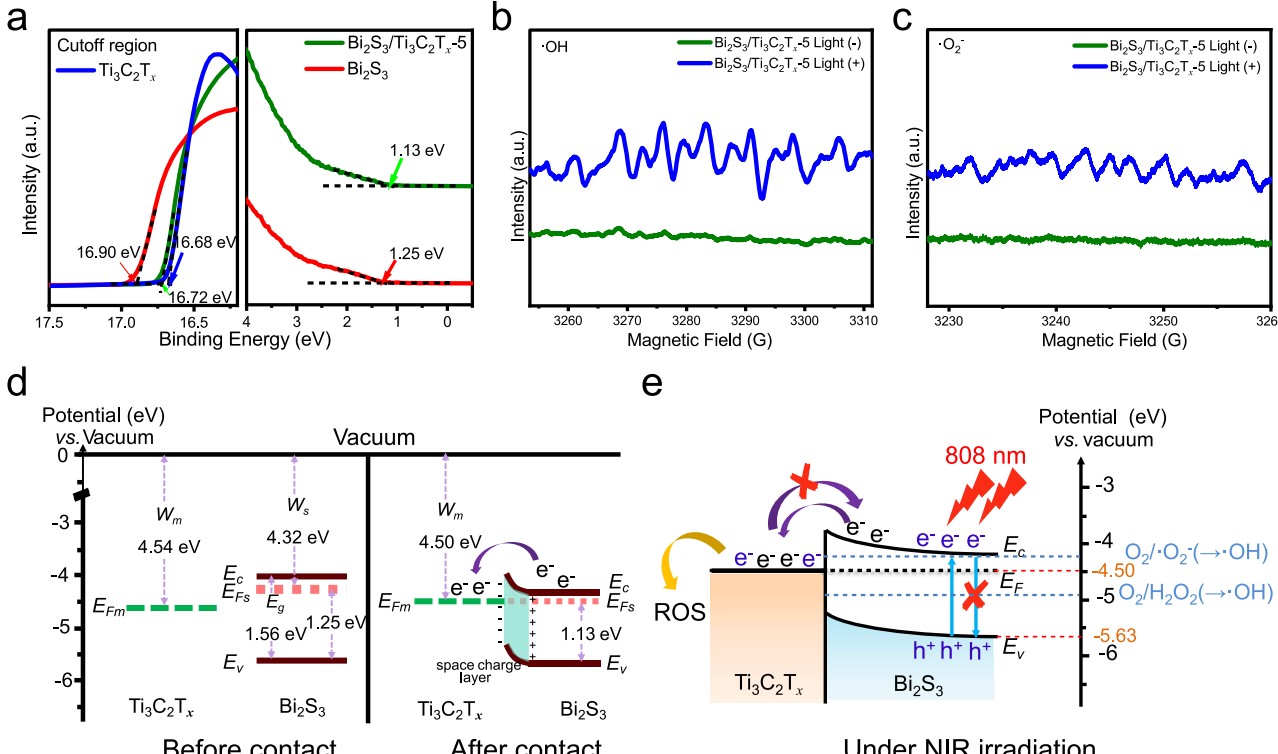

**Fig. 3 The mechanism of interfacial engineering based on work function.** **a** UPS spectra measured by He I ($h\nu = 21.22$ eV) spectra the secondary electron cutoff of Ti$_3$C$_2$T$_x$ and valence band of Bi$_2$S$_3$ and Bi$_2$S$_3$/Ti$_3$C$_2$T$_x$-5 Schottky catalyst with respect to the Fermi level ($E_F$). **b** ESR spectra of hydroxyl radicals of Bi$_2$S$_3$/Ti$_3$C$_2$T$_x$-5. **c** ESR spectra of superoxide radical of Bi$_2$S$_3$/Ti$_3$C$_2$T$_x$-5. **d** Energy scheme before and after contact of Ti$_3$C$_2$T$_x$ and $n$-type Bi$_2$S$_3$. $W$, work function. $E_g$, energy gap. $E_c$, conduction band minimum, $E_v$, valence band maximum. **e** The mechanism for the enhanced the yield of ROS via NIR-induced progress based on the Schottky heterostructure of Bi$_2$S$_3$/Ti$_3$C$_2$T$_x$. Source data are provided as a Source Data file.

formation of an electron depletion zone and an upward bend in the band edge. The negatively charged layer of electrons accumulated on the $Ti_3C_2T_x$ surface. Thus, the Schottky barrier and a built-in electric field were formed at the interface of the Schottky heterojunction[40,41]. The obtained energy scheme appears in Fig. 3d. When irradiated by NIR (Fig. 3e), the photogenerated electrons from $Bi_2S_3$ leaped into the conduction band and rapidly migrated to the $Ti_3C_2T_x$ surface. The Schottky barrier nearly impeded the backflow of electrons, which efficiently separated the photogenerated electron–hole pairs and increased the free carrier concentration. As a result, more photogenerated charges from $Bi_2S_3$ under NIR light were rapidly transferred to $Ti_2C_3T_x$ and then captured by surrounding species to form ROS. Moreover, the intimate contact between $Ti_3C_2T_x$ and $Bi_2S_3$ provided a transport channel for the photogenerated carriers. Therefore, the optimum matching of $Bi_2S_3/Ti_3C_2T_x$ produced more ROS.

To illustrate the role of $Ti_3C_2T_x$ in the charge transfer, a DFT calculation was performed. The DFT calculation of $Ti_3C_2F_2$, $Ti_3C_2O_2$, $Ti_3C_2(OH)_2$, and $Bi_2S_3$ confirmed $Ti_3C_2T_x$ and $Bi_2S_3$ can also form a Schottky junction, theoretically (Supplementary Figs. 17 and 18). The structure of the $Ti_3C_2T_x$ slab is exhibited in Supplementary Fig. 19. Since the content of F-terminated and O-terminated $Ti_3C_2T_x$ occupied most of $Ti_3C_2T_x$ according to the result of XPS, the DFT calculations were performed at the $Bi_2S_3/Ti_3C_2F_2$ and $Bi_2S_3/Ti_3C_2O_2$ interface (Fig. 4). WF is 4.787 eV for $Bi_2S_3/Ti_3C_2F_2$ and 5.867 eV for $Bi_2S_3/Ti_3C_2O_2$ and the electrostatic potential of $Bi_2S_3$ is higher than that of $Ti_3C_2F_2$ or $Ti_3C_2O_2$, resulting in the formation of an interfacial electrical field and Schottky barrier (Fig. 4a, b). These results indicated that the electrons would be driven to move from $Bi_2S_3$ to $Ti_3C_2F_2$ or $Ti_3C_2O_2$ due to the difference of $E_F$, inevitably resulting in an increased electron concentration on the side of $Ti_3C_2T_x$ and a decreased electron density on the side of $Bi_2S_3$. The differential charge densities of $Bi_2S_3/Ti_3C_2F_2$ and $Bi_2S_3/Ti_3C_2O_2$ also intuitively confirmed this result. A distinct redistribution of electrons occurred on the heterointerface (Fig. 4c, d); abundant electrons gathered at the bottom of $Ti_3C_2F_2$ (Fig. 4c) or $Ti_3C_2O_2$ (Fig. 4d), while the top of $Bi_2S_3$ became an electron-deficient region. Most electrons gathered on the side of $Ti_3C_2T_x$, further confirming that they were transferred from $Bi_2S_3$ to $Ti_3C_2T_x$. Thus, this junction could effectively regulate the interfacial charge transfer behaviors[42]. To better explain this regulatory effect, the contributions of $Ti_3C_2F_2$ and $Ti_3C_2O_2$ to the electronic band gap structure of $Bi_2S_3$ are simulated in Fig. 4e, f. These results explicitly demonstrate that the introduction of $Ti_3C_2T_x$ induced the formation of the impurity levels in the band gap of $Bi_2S_3$, accelerating the generation of electrons and reducing the band gap of $Bi_2S_3$. In reality, the theoretical band gaps of $Bi_2S_3/Ti_3C_2F_2$ and $Bi_2S_3/Ti_3C_2O_2$ are 1.23 and 1.29 eV, respectively, much lower than that of $Bi_2S_3$. The reduced band gap was beneficial to the production of more photogenerated electrons under the same irradiation, thus promoting the ROS yield.

Accordingly, the following mechanism is proposed and schematically illustrated in Fig. 4g. Pristine $Ti_3C_2T_x$ only exhibits low photothermal properties due to its LSPR effect under 808 nm NIR light irradiation. $Bi_2S_3$ only produces inferior heat and fewer ROS, because few photogenerated electrons can be excited by NIR light with its relatively wide energy gap. Given the rapid recombination of carriers, only a few electrons can be captured and converted to ROS. Due to the deep-level defects in $Bi_2S_3$, sulfur vacancies and $Bi_S$ antisite (Bi replacing S) defects become electron–hole nonradiative recombination centers to produce low-density phonons, inducing sluggish temperature change[43,44]. After contact between $Ti_3C_2T_x$ and $Bi_2S_3$, the interaction of the energy band and interfacial structure promote photothermal and

photodynamic effects. A Schottky junction can be formed at the $Bi_2S_3/Ti_3C_2T_x$ interface due to their different WF values. The irreversible charge migration from $Bi_2S_3$ to $Ti_3C_2T_x$ can occur until equilibrium is reached. Moreover, as the energy gap of $Bi_2S_3$ decreases after combination with $Ti_3C_2T_x$, more excited electrons can leap into the conduction band. As a result, under the same light irradiation, more photogenerated electrons induced by $Bi_2S_3$ can be rapidly transferred to $Ti_3C_2T_x$. Then, the declining interfacial impedance can be beneficial to the transition of photoexcited charges, thus decreasing the recombination of the photoinduced electrons and holes. In this heterojunction, given its excellent conductivity, $Ti_3C_2T_x$ has the role of an electron trapper, which not only suppresses the carrier recombination on the surface of $Bi_2S_3$, but also promotes the charge separation and transfer, thus inducing higher ROS yields. The NIR-induced LSPR effect of $Ti_3C_2T_x$ may also lead to the injection of hot electrons, favoring the separation of the photogenerated carriers[45,46]. Simultaneously, more hot carriers can also give rise to crystal lattice vibrations, further elevating the temperature. This sufficient contact of $Ti_3C_2T_x$ and $Bi_2S_3$ minimizes heat loss and improves heat utilization. Moreover, the homogeneous distribution of $Bi_2S_3$ on the surface of $Ti_3C_2T_x$ is beneficial for rapid and uniform heating. Hence, the interaction between $Ti_3C_2T_x$ and $Bi_2S_3$ improves the photothermal and photodynamic performance.

**Antibacterial assessment in vitro**. Two typical pathogenic bacteria, *Staphylococcus aureus* (*S. aureus*; Gram positive) and *Escherichia coli* (*E. coli*; Gram negative) were chosen to evaluate the antibacterial performance of the materials. After 10 min of NIR irradiation, 200 ppm of $Bi_2S_3/Ti_3C_2T_x$-5 exhibited effective bacteria-killing efficiencies of 99.86% and 99.92% against *S. aureus* and *E. coli*, respectively. The experiments were conducted by spread plate, live/dead fluorescence staining, and SEM antibacterial images (Supplementary Figs. 20–23). The permeability of the bacterial cell membrane measured by *ortho*-nitrophenyl-β-galactoside hydrolysis revealed the antibacterial mechanism against both *S. aureus* and *E. coli*. As shown in Fig. 5a, b, no significant improvement was observed for the optical density at 420 nm ($OD_{420}$) values of *S. aureus* and *E. coli* for $Ti_3C_2T_x$ compared to the control group, indicating unchanged bacterial membrane permeability. The highest $OD_{420}$ value (for $Bi_2S_3/Ti_3C_2T_x$-5) suggested that the bacterial membranes had been irreversibly damaged. These results accorded well with the bacterial morphological evaluation using SEM (Supplementary Fig. 23). The antibacterial efficacy shown in Supplementary Fig. 24 indicated that the synergy of ROS and photothermal effect achieved much better antibacterial performance than single ROS or photothermal effect alone. Additionally, the samples in the dark for 12 h did not exhibit antibacterial performance (Supplementary Fig. 25). The antibacterial performance of materials under 660 nm light or simulated solar irradiation was also assessed. As shown in Supplementary Figs. 26 and 27, $Bi_2S_3/Ti_3C_2T_x$-5 exhibited inferior antibacterial performance under these conditions than that under 808 nm light irradiation.

In summary, none of the as-prepared materials showed antibacterial properties for ten minutes under static conditions. The morphologies, compositions, and structures of these materials had negligible effects on bacterial viability. However, $Bi_2S_3/Ti_3C_2T_x$-5 showed the best antibacterial efficacy after only 10 min of NIR light irradiation. The cell viability results showed that all the as-prepared samples were not cytotoxic, and that only a short irradiation time could decrease the cell proliferation (Supplementary Fig. 28). Although the recent silver/MXene nanosheets reported by Pandey et al.[18] exhibited high

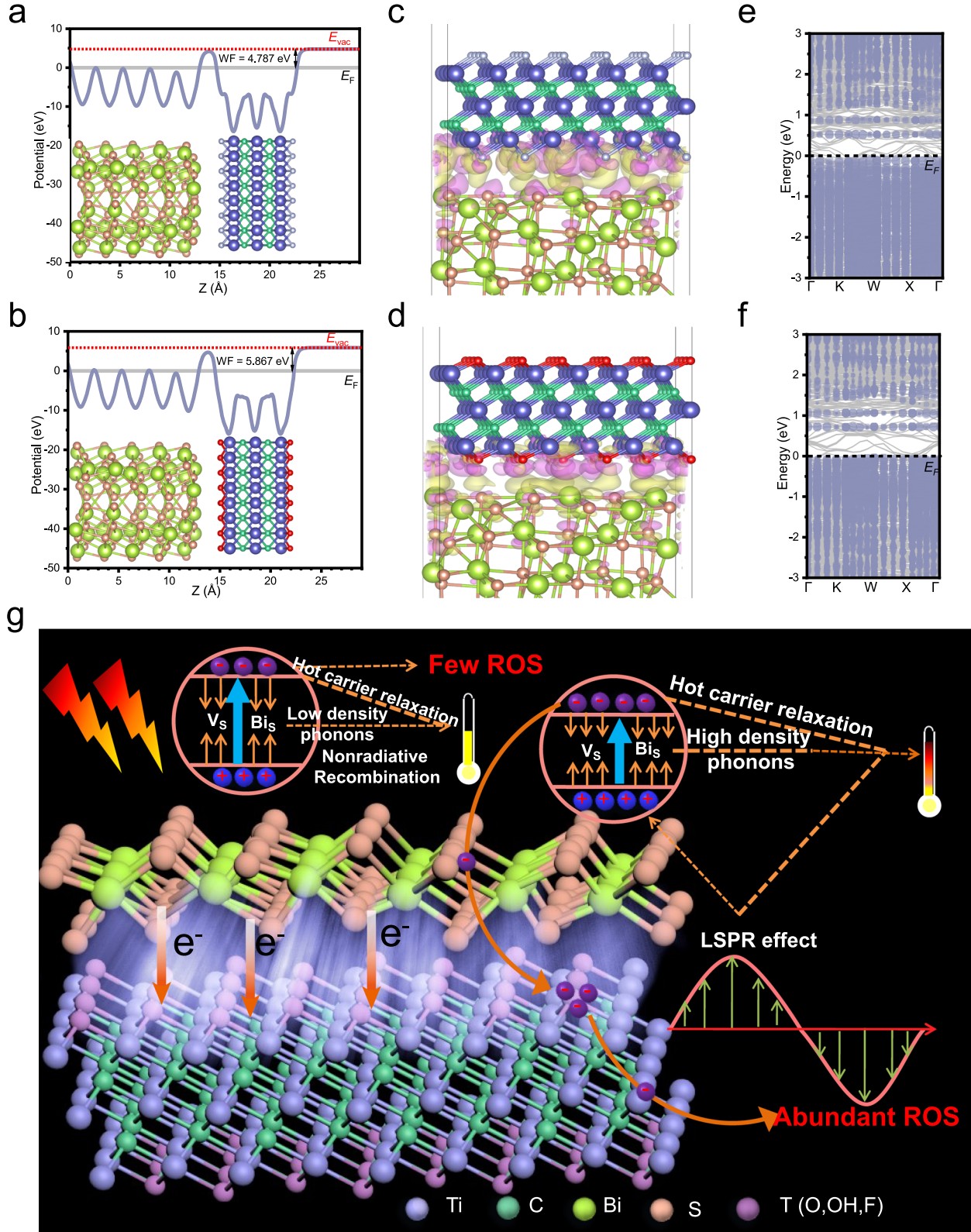

**Fig. 4 DFT calculation of the Bi$_2$S$_3$/Ti$_3$C$_2$F$_2$ and Bi$_2$S$_3$/Ti$_3$C$_2$O$_2$ structure. a, b** Electrostatic potential along $z$ axis of Bi$_2$S$_3$/Ti$_3$C$_2$F$_2$ (**a**) and Bi$_2$S$_3$/Ti$_3$C$_2$O$_2$ (**b**) interface. The red dashed lines denote vacuum level ($E_{vac}$), the gray lines represent Fermi level ($E_F$). **c, d** The differential charge density of Bi$_2$S$_3$/Ti$_3$C$_2$F$_2$ (**c**) and Bi$_2$S$_3$/Ti$_3$C$_2$O$_2$ (**d**) interface. Pink and yellow colors denote electron excess and deficiency area, respectively. **e, f** Electronic band gap structure of Bi$_2$S$_3$/Ti$_3$C$_2$F$_2$ and Bi$_2$S$_3$/Ti$_3$C$_2$O$_2$. Blue-gray colors denote contributions from the bulk part of the Ti$_3$C$_2$F$_2$ (**e**) and Ti$_3$C$_2$O$_2$ (**f**). The darkness of this color is determined by the degree of contribution. **g** Schematic diagram for photodynamic and photothermal mechanism between Ti$_3$C$_2$T$_x$ and Bi$_2$S$_3$. Source data are provided as a Source Data file.

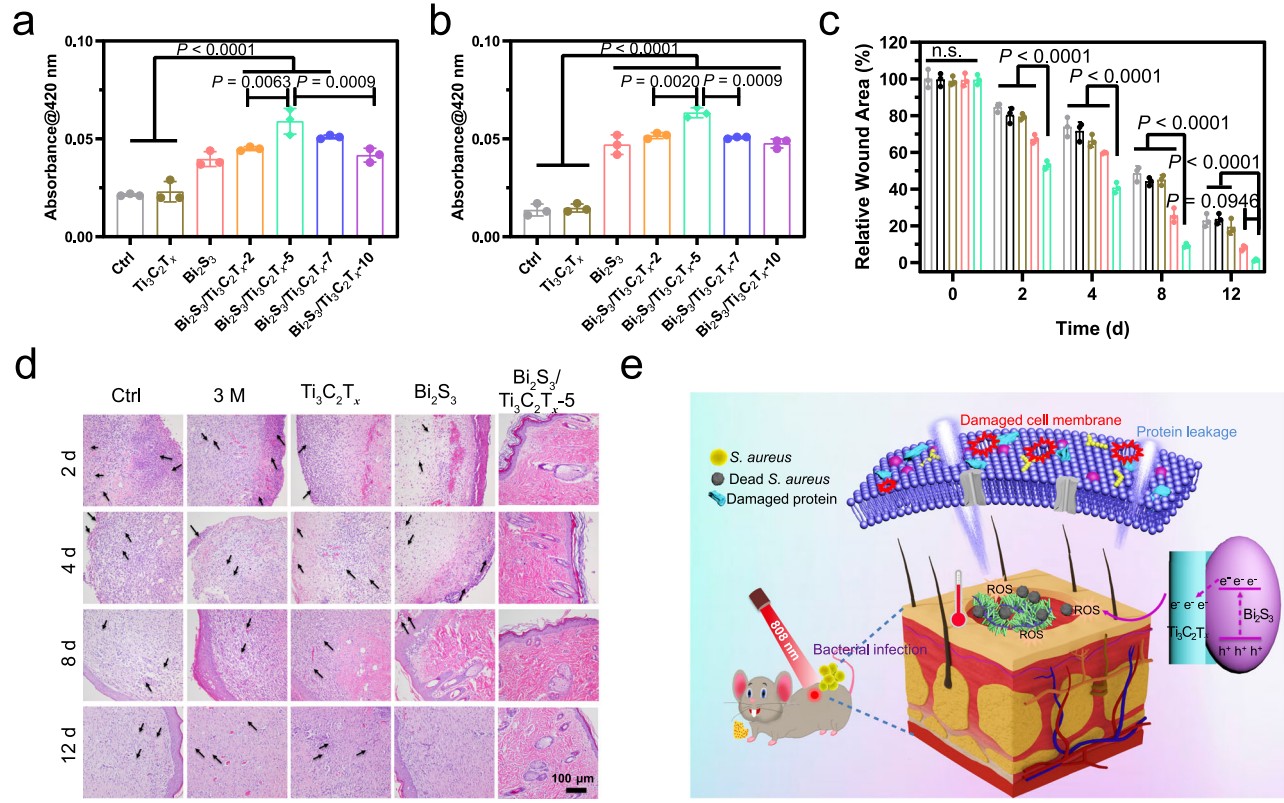

**Fig. 5 The antibacterial property in vitro and in vivo. a, b** ONPG hydrolysis of *S. aureus* (**a**) and *E. coli* (**b**) after 10 min light of $Ti_3C_2T_x$, $Bi_2S_3$, $Bi_2S_3/Ti_3C_2T_x$-2, $Bi_2S_3/Ti_3C_2T_x$-5, $Bi_2S_3/Ti_3C_2T_x$-7 and $Bi_2S_3/Ti_3C_2T_x$-10 in vitro. **c** The Quantitative result of relative wound area on 2, 4, 8, and 12 d of control, 3 M wound dressing, $Ti_3C_2T_x$, $Bi_2S_3$, and $Bi_2S_3/Ti_3C_2T_x$-5 on *S. aureus* (10 μL, $1 \times 10^8$ CFU/mL) to infective wound healing in vivo. Gray circles indicate the group of Ctrl, black circles indicate the group of 3 M, brown circles indicate the group of $Ti_3C_2T_x$, pink circles indicate the group of $Bi_2S_3$, and green circles indicate the group of $Bi_2S_3/Ti_3C_2T_x$-5. **d** Histological images of wound tissue collected on 2, 4, 8, and 12 d with hematoxylin and eosin (H&E) staining. **e** The Schematic illustration of $Bi_2S_3/Ti_3C_2T_x$ Schottky catalyst and antibacterial mechanism diagram of $Bi_2S_3/Ti_3C_2T_x$ under 808 nm irradiation. Data are presented as mean ± standard deviations from a representative experiment (*n* = 3 independent samples). *P*-values were analyzed by one-way ANOVA with Tukey's multiple comparisons post hoc test. Source data are provided as a Source Data file.

antibacterial efficacy of more than 99% of *E. coli*, as compared with the current study (Supplementary Table 2), it was achieved at a very low bacterial concentration of $2*10^3$ with much longer time of 24 h while in this work, $Bi_2S_3/Ti_3C_2T_x$ only took 10 min to kill 99.86% of *S. aureus* and 99.92% of *E. coli* with a concentration of $10^7$ under 808 nm NIR light irradiation. Furthermore, the bacteria were killed by released Ag and the physical action of MXene nanosheets in the former, which exhibited potential toxicity from released Ag. It has been reported that long time contact with bacteria can also induce bacterial resistance, even for Ag[47]. In contrast, $Bi_2S_3/Ti_3C_2T_x$ killed bacteria rapidly through a synergy of ROS and hyperthermia under 808 nm light irradiation with good cytocompatibility simultaneously.

Compared to other 2D nanomaterials of black phosphorous, $MoS_2$, graphene oxide, carbon nitride, and Ag-based materials, MXene has excellent metallic conductivity, surface hydrophilicity, and photothermal conversion property. Thus, the antibacterial activity of other MXene-based materials was also assessed. As shown in Supplementary Fig. 29, similar antibacterial activity was observed $Bi_2S_3/V_2CT_x$-5 and $Bi_2S_3/Nb_2CT_x$-5. This exciting finding definitely exhibited an enormous potential in coupling with $Bi_2S_3$ with a wide variety of MXene as effective antibacterial materials.

**Antibacterial and tissue reconstruction in vivo**. Based on the aforementioned in vitro results, a rat wound model with *S. aureus* ($1 \times 10^8$ CFU $mL^{-1}$) was built to evaluate the in vivo therapeutic

efficacies with gauze, a 3 M wound dressing, $Bi_2S_3/Ti_3C_2T_x$-5, and a control group. The calculated quantitative relative wound areas were shown in Fig. 5c; the wounds of the $Bi_2S_3/Ti_3C_2T_x$-5 group almost healed up, illustrating a clear wound healing trend. The visual results can be seen in Supplementary Fig. 31. Further, hematoxylin and eosin (H&E) staining of wound tissues collected on 2, 4, 8, and 12 d confirmed the wound healing activity (Fig. 5d). Conversely, a high number of inflammatory cells appeared for the control, 3 M, $Ti_2C_3$, and $Bi_2S_3$ groups after the 2-d treatment. This phenomenon did not occur in the $Bi_2S_3/Ti_3C_2T_x$-5 group. Even on day 12, some inflammatory cells were visualized in the control, 3 M, and $Ti_3C_2T_x$ groups, indicating that the wound still had not completely healed. The data demonstrated that the $Bi_2S_3/Ti_3C_2T_x$ Schottky junction significantly promoted wound healing under 808 nm NIR irradiation.

Accordingly, the antibacterial mechanism is schematically illustrated in Fig. 5e. The optimized $Bi_2S_3/Ti_3C_2T_x$ Schottky junction produced more electrons and effectively separated electrons and holes, causing irreversible damage to bacteria. The photothermal effect of the samples decreased the viability of the bacteria. The synergistic impacts of the ROS and photo-thermal effect killed the bacteria rapidly and effectively. The synthesized $Bi_2S_3/Ti_3C_2T_x$-5 exhibited excellent cytocompatibility and biocompatibility, which promoted the formation of collagen fibers (Supplementary Fig. 32), thus accelerating wound healing. Most importantly, this Schottky junction showed excellent in vivo biosafety without damaging the main organs (i.e., the heart, liver, spleen, lung, and kidneys) (Supplementary Fig. 33).

## Discussion

A Schottky junction of $Bi_2S_3/Ti_3C_2T_x$ with a built-in electric field was successfully prepared. It demonstrated highly effective photoresponsive bacteria-killing efficacies of 99.86% and 99.92% against *S. aureus* and *E. coli*, respectively, under 10 min of 808 nm NIR light irradiation. The underlying mechanisms involved enhanced photocatalytic and photothermal properties, which originated from the rapid transfer of photogenerated charges at the interface, the deep-level defects of $Bi_2S_3$, and the LSPR of $Ti_3C_2T_x$ after tight contact. The difference in the work functions of $Ti_3C_2T_x$ and $Bi_2S_3$ caused the contact potential difference, which forced the transfer of the photogenerated charges across the interface and increased the local electron density on $Ti_3C_2T_x$, reducing the recombination of the electron–hole pairs. We utilize the work function difference between components to design a biocompatible Schottky junction with highly effective NIR responsive bactericidal ability through enhanced photocatalytic and photothermal properties. This study offers fresh insights into designing photoelectrical devices for specific applications using two or more components with different work functions.

## Methods

**Preparation of $Ti_3C_2T_x$ nanosheets**. Hydrogen fluoride solution and delamination were used to synthesize $Ti_3C_2T_x$ nanosheets from $Ti_3AlC_2$[13]. Specifically, 1.0 g of $Ti_3AlC_2$ was gently added in 20 mL of hydrogen fluoride solution (40 wt.%) for 15 min using magnetic stirring to alleviate the exothermic reaction. After etching for 24 h at 40 °C, this mixture was washed and centrifuged with deionized water several times until the pH value of the supernatant equaled or exceeded 6. Then, the multilayer $Ti_3C_2T_x$ was dried under vacuum at 30 °C. The dried precipitation was dispersed in 20 mL of tetrapropylammonium hydroxide (25 wt.%) under stirring for 3 d, followed by centrifugation and thorough washing three times with deionized water. Thereafter, an aqueous solution containing MXene nanosheets was obtained using an ultrasonic cell grinder in ice water for 1 h and centrifugation at $2150 \times g$. The supernatant was stored in the refrigerator for further experiments. The concentration of the $Ti_3C_2T_x$ nanosheets was 1 mg mL$^{-1}$.

**Synthesis of $Bi_2S_3/Ti_3C_2T_x$**. To synthesize the $Bi_2S_3/Ti_3C_2T_x$ composites, $Ti_3C_2T_x$ aqueous solution (pH = 6.42) was sonicated by an ultrasonic cell grinder. This process was under an ice bath for 2 h at 20% amplitude in pulsed mode with 2 s on 2 s off. And then 0.2 mM of $Bi(NO_3)_3\cdot5H_2O$ was respectively dissolved in deionized water containing a 1, 2.5, 3.5, and 5 mL of $Ti_3C_2T_x$ aqueous solution and stirred for 2 h to ensure enough time for electrostatic adsorption. Subsequently, 27 mg of thioacetamide ($C_2H_5NS$) was added and stirred for 30 min. The mixed solution was transferred into a 50 mL Teflon-lined autoclave to maintain 160 °C for 8 h. After cooling down, the final black precipitate was rinsed and centrifuged with deionized water, and dried in a vacuum drying oven. The mass ratios of $Ti_3C_2T_x$ were 2%, 5%, 7%, and 10% of the mass of $Bi_2S_3$, and the obtained samples were labeled as $Bi_2S_3/Ti_3C_2T_x$-2, $Bi_2S_3/Ti_3C_2T_x$-5, $Bi_2S_3/Ti_3C_2T_x$-7, and $Bi_2S_3/Ti_3C_2T_x$-10, respectively. The preparation of $Bi_2S_3$ was similar to that of $Bi_2S_3/Ti_3C_2T_x$ without the introduction of $Ti_3C_2T_x$.

**Characterization**. The morphologies and microstructures of all the samples were analyzed by scanning electron microscopy (JSM-7800F, Japan) and transmission electron microscopy (JEOL-2100F, Japan). X-ray diffraction was performed on a Bruker diffractometer (D8 Advance, Germany) with Cu Kα radiation for $2\theta$ from 3° to 70°. X-ray photoelectron spectroscopy (ThermoFisher Scientific 250Xi, USA) of the samples was conducted for elemental analysis. All the binding energies were calibrated by the C 1$s$ peaks located at 284.8 eV. The zeta potential and hydrodynamic sizes were measured by dynamic light scattering analysis (90 Plus/BI-MAS, Brookhaven, USA). Raman spectroscopy was carried out using the 532 nm solid-state laser to verify the MXene sheets in the polymer. Ultraviolet-visible-NIR absorption spectra and diffuse reflectance absorption spectra were employed using a UV-2700 spectrophotometer (Shimadzu). Photoluminescence (PL) spectra were measured by a Fluorolog-3 fluorescence spectrophotometer under 325 nm excitation. Ultraviolet photoemission spectroscopy was conducted on an Escalab 250Xi spectrometer using He I resonance lines (21.22 eV).

**Measurement of photothermal effects**. The photothermal effects of all the samples ($Ti_3C_2T_x$, $Bi_2S_3$, $Bi_2S_3/Ti_3C_2T_x$-2, $Bi_2S_3/Ti_3C_2T_x$-5, $Bi_2S_3/Ti_3C_2T_x$-7, and $Bi_2S_3/Ti_3C_2T_x$-10) were analyzed with a thermal imager. Each sample (concentration: 200 ppm) was irradiated in 96-well plates by an 808 nm laser (0.7 W cm$^{-1}$) for 10 min. The variations in temperatures and thermal images were recorded by a thermal camera (FLIR, E50) at 1-min intervals. Also, 200 ppm of $Bi_2S_3/Ti_3C_2T_x$-5 was exposed under the same condition for 10 min and then cooled

in the dark for 15 min. Four cycles were processed and the heating-cooling curve was recorded every 2 min. The photothermal conversion efficiency ($\eta$) of $Bi_2S_3/Ti_3C_2T_x$-5 was calculated using Eq. (1).

$$\eta = \frac{hS(T_{max} - T_0) - Q_0}{W(1 - 10^{-A})} \qquad (1)$$

where $\eta$ represents the heat transfer coefficient of $Bi_2S_3/Ti_3C_2T_x$-5, $S$ denotes the heated area, $T_{max}$ is the highest temperature of $Bi_2S_3/Ti_3C_2T_x$-5, $T_0$ is the initial temperature, $Q_0$ indicates the heating absorption energy of container, $W$ is the power source for the 808 nm NIR light irradiation, and $A$ is the absorbance of $Bi_2S_3/Ti_3C_2T_x$-5 at 808 nm.

Moreover, Eq. (2) was used.

$$hS = \frac{m_{H_2O}C_{H_2O}}{\tau_s} \qquad (2)$$

where $m_{H_2O}$ represents the total mass of $H_2O$, $C_{H_2O}$ denotes the specific heat capacity of $H_2O$, and $\tau_s$ is the time constant quantity of $Bi_2S_3/Ti_3C_2T_x$-5.

Then, Eq. (3) was used for the cooling period (i.e., after removing the 808 nm light source).

$$t = -\tau_s \times \ln\theta = -\tau_s \ln\frac{T - T_0}{T_{max} - T_0} \qquad (3)$$

where $T_{max}$ is the highest temperature of $Bi_2S_3/Ti_3C_2T_x$-5, $T_0$ is the initial temperature, $T$ represents the real-time temperature. Thus, the time constant quantity was calculated from the linear regression curve in the cooling period of $Bi_2S_3/Ti_3C_2T_x$-5.

### Measurement of photodynamic effects

*Detection of ROS*. The amounts of ROS generated from the solutions of the samples under 808 nm NIR light were obtained using a Reactive Oxygen Species Assay Kit. 2′,7′-dichlorofluorescein diacetate can be converted into 2′,7′-dichlorofluorescein (DCF) by reaction with ROS to generate fluorescence, which can be detected by a microplate reader. The liquid was measured every 20 s, and every sample was tested in triplicate to verify the result. Further, the electron spin resonance spectra were performed by a JES-FA200 spectrometer. The electron spin resonance spectra were measured under 808 nm NIR light. The superoxide radicals ($\cdot O_2^-$) were measured by Nitro Blue Tetrazolium (NBT) after 808 NIR light irradiation. The amount of $\cdot O_2^-$ can be examined by the absorption spectra of monoformazan (MF) because MF is produced through the reaction between $\cdot O_2^-$ and NBT dissolved in dimethyl sulfoxide (DMSO) ($2.5 \times 10^{-5}$ mol L$^{-1}$) solution. 1,3-diphenylisobenzofuran (DPBF) was used to detect the production of the $^1O_2$ during light irradiation. The absorption of DPBF at 416 nm was decreased when reacting with $^1O_2$.

*Electrochemical measurement*. The electrochemical testing of the samples was undertaken using an electrochemical workstation (CHI660E, China) with a conventional three-electrode system, where an Ag/AgCl electrode served as the reference electrode, a platinum foil was the counter electrode, and each sample was used as the working electrode. To fabricate the working electrode, 200 μL of the mixture, each containing 4 mg of the sample, 1 mL of deionized water, and 80 μL of Nafion solution, were dropped on fluoride-tin oxide conductor glass to form a uniform film. All measurements were processed in 0.1 M $Na_2SO_4$ solution at 25 °C. The photocurrents were recorded under 808 nm NIR irradiation. The electrochemical impedance spectroscopy was measured from 100 kHz to 0.01 Hz, with an amplitude of 10 mV, with and without NIR irradiation. The transient photocurrent responses were collected with and without the NIR light.

*Density function theory calculation*. To illustrate the interface properties of $Bi_2S_3$-$Ti_3C_2T_2$ (T = F, O, and OH), first-principle calculations were performed using the Vienna Ab initio Simulation Package (VASP 5.3.5). The valence–core electron interactions were treated with projector-augmented wave potentials, and the electron exchange–correlation interactions were described by the generalized gradient approximation with the Perdew–Burke–Emzerh functional. Considering the long-range interaction at the interface, van der Waals interactions were considered using the density function theory-D3 correlation. The energy cutoff for the plane-wave basis was set to 500 eV. The k points were sampled with a $2 \times 2 \times 1$ k-point grid using the Monkhorst–Pack method.

**Antibacterial assessment in vitro**. The antimicrobial measurements against *E. coli* (ATCC 8099) and *S. aureus* (ATCC 25923) were conducted using $10^7$ colony-forming unit (CFU) mL$^{-1}$. The bacteria were cultured in Luria-Bertani culture medium at 37 °C. For the spread plate method, 200 μL of bacterial suspension was added to 96-well plates containing different samples. The final concentration was 200 ppm. After 808 nm irradiation (0.7 W cm$^{-1}$) for 10 min, the bacteria were cultured on agar plates at 37 °C. Moreover, without light irradiation was used for the control group. The antibacterial efficiency of every sample was calculated by the

number of bacterial colonies on the plates using Eq. (4).

$$\text{Antibacterial ratio}(\%) = \frac{\text{CFU}_{\text{Control}} - \text{CFU}_{\text{Sample}}}{\text{CFU}_{\text{Control}}} \times 100\% \qquad (4)$$

The antibacterial performance of materials under 660 nm light or simulated solar irradiation was also examined by the spread plate method. The $10^7$ CFU mL$^{-1}$ each of *E. coli* and *S. aureus* in the dark were suspended in samples (200 ppm) and cultured under continuous shaking at 120 r.p.m. (rotational radius of 10 cm) and a constant temperature of 37 °C for 12 h to assess the antibacterial performance. The antibacterial activity of ROS alone was assessed in a water bath at a constant temperature of 25 °C. The antibacterial activity of the photothermal effect alone was measured by the addition of ascorbic acid. Laser scanning confocal microscopy (CLSM) was utilized to evaluate the antibacterial performance. After interactions with different samples for 10 min with and without the 808 nm light, the bacteria were stained with the live/dead Baclight bacterial viability kit (SYTO9 dye and PI dye), and then washed thrice with phosphate-buffered saline (PBS) and observed by CLSM. To analyze the bacterial morphologies, the *E. coli* and *S. aureus* in the 96-well plates were fixed with a 2.5% glutaraldehyde solution for 40 min and dehydrated using graded ethanol solutions (20, 40, 60, 80, and 100%) for 15 min. After drying, the morphologies and microstructures were observed by scanning electron microscopy.

Bacterial membrane permeability assay: 2-nitrophenyl-β-galactoside was used to measure the change in bacterial membrane permeability. In brief, the bacteria were cultured using isopropyl β-d-1-thiogalactopyranoside medium at 37 °C. Subsequently, the bacteria were diluted with sterile PBS until the optical density (OD) was 0.05–0.1. Then, 200 ppm of Ti$_3$C$_2$T$_x$, Bi$_2$S$_3$, Bi$_2$S$_3$/Ti$_3$C$_2$T$_x$-2, Bi$_2$S$_3$/Ti$_3$C$_2$T$_x$-5, Bi$_2$S$_3$/Ti$_3$C$_2$T$_x$-7, Bi$_2$S$_3$/Ti$_3$C$_2$T$_x$-10, and the bacterial suspensions were irradiated with 808 nm (0.7 W cm$^{-1}$) light for 10 min. The supernatant absorbance was measured at 420 nm.

**Biocompatibility assessment in vitro**
*Cell culture*. NIH-3T3 cells (ATCC CRL-1658) were cultured in a cell culture flask with α minimum essential medium (α-MEM, Gibco) including 10% fetal bovine serum and 1% penicillin–streptomycin solution in a humidified atmosphere incubator with 5% CO$_2$ at 37 °C.

*Cell spreading assay*. Cell adhesion experiments were conducted to visualize the actin cytoskeleton and nucleus using fluorescein isothiocyanate-phalloidin and 4,6-diamidino-2-phenylindole, respectively. The cells were seeded in a 96-well plate at a cell density of $2.5 \times 10^4$ cells·cm$^{-2}$. After culturing for 24 h, 200 ppm of each sample (Ti$_3$C$_2$T$_x$, Bi$_2$S$_3$, Bi$_2$S$_3$/Ti$_3$C$_2$T$_x$-2, Bi$_2$S$_3$/Ti$_3$C$_2$T$_x$-5, Bi$_2$S$_3$/Ti$_3$C$_2$T$_x$-7, and Bi$_2$S$_3$/Ti$_3$C$_2$T$_x$-10) and cell culture medium were used to co-culture the cells. After 24 h, cells were fixed with 4% formaldehyde and rinsed again with PBS. The cells were then stained with fluorescein isothiocyanate in darkness for 40 min, and then washed thrice with PBS. Further, the cells were stained with 4, 6-diamidino-2-phenylindole for 30 s in every well. The F-actin and nuclei of cells were observed by CLSM.

*Cell proliferation assay*. The cell viability and proliferation were evaluated quantitatively by a 3-(4, 5-dimethylthiazol-2-yl)-2, 5-diphenyl tetrazolium bromide (MTT) assay. After co-culturing with the cell culture medium and the samples for 1 and 3 d, the supernatant was discarded, and 90 μL of cell culture medium and 10 μL of MTT solution were added. After culturing for 4 h, the supernatant was removed and 110 μL of dimethyl sulfoxide was used to dissolve formazan. The optical densities of the dimethyl sulfoxide solutions were measured with a microplate reader at a wavelength of 490 nm. The data were analyzed by using OriginPro 8.5, Origin 8, Microsoft Excel 2013, and GraphPad Prism 8.0.

**In vivo wound healing test**. Male Wistar rats (10–12 weeks old, each ~350 g in body weight) were bought from the Beijing Huafukang Biotechnology Company. The study was carried out in accordance with the Guide for the Care and Use of Laboratory Animals of the National Institutes of Health. The ethical aspects of the animal experiments were approved by the Animal Ethical and Welfare Committee (AEWC) of the Institute of Radiation Medicine, Chinese Academy of Medical Sciences. Five groups with three rats in every group were used to experiments, namely, Control, standard 3M wound dressing, Ti$_3$C$_2$T$_x$, Bi$_2$S$_3$, and Bi$_2$S$_3$/Ti$_3$C$_2$T$_x$-5 group. After anesthesia, Except for the control group, 10 μL of *S. aureus* ($1 \times 10^8$ CFU mL$^{-1}$) was added to wounds with a diameter of 16 mm to establish the infection model. Then, 10 μL of each sample was added to the wounds and irradiated under 808 nm NIR light for 10 min. The wounds were photographed and sacrificed to use for histological staining, namely hematoxylin and eosin, Sirius red staining, and Masson staining. Moreover, the heart, liver, spleen, lung, and kidneys were removed from each mouse and stained with hematoxylin and eosin after 12 d of treatment. Image J (1.51j8) and CaseViewer 2.3 was used to deal with image presentation and quantification.

**Statistics and reproducibility**. All the quantitative data in each experiment were evaluated and analyzed by one-way or two-way analysis of variance and expressed as the mean values ± standard deviations, followed by Tukey's multiple

comparisons post hoc test to evaluate the statistical significance of the variance. The n.s. present $P > 0.05$ and ****$P < 0.0001$ were considered statistically significant.

**Reporting summary**. Further information on research design is available in the Nature Research Reporting Summary linked to this article.

## Data availability
All other data are available from the corresponding author upon reasonable requests. Source data are provided with this paper.

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

## Acknowledgements

This work is jointly supported by the National Science Fund for Distinguished Young Scholars 51925104, National Natural Science Foundation of China nos. 51871162, 51671081, and 81870809, Natural Science Fund of Hubei Province, 2018CFA064, RGC/ NSFC (N_HKU725-1616), Hong Kong ITC (ITS/287/17, GHX/002/14SZ), as well as Health and Medical Research Fund (No. 03142446).

## Author contributions

J.L., X.L., and S.W. conceived and designed the concept of the experiments. J.L. performed the experiments and conducted the material characterizations. J.L., X.L., and S.W. analyzed the experimental data and co-wrote the manuscript. Y.Z, W.H., Y.Q., T.Z., and X.W. helped with the data analysis of the mechanism. C.L, K.W. K.Y., Z.C., Y.L., Z.L., S.Z., and X.W. provided important experimental insights and performed data analysis partially. All the authors discussed, commented, and agree on the manuscript.

## Competing interests

The authors declare no competing interests.
