## [Peer Review File · Nature Communications]

REVIEWER COMMENTS

Reviewer #1 (Remarks to the Author):

This manuscript describes a study on the synthesis and antibacterial activity of a Ti₃C₂/MXene/Bi₂S₃ hybrid structure. Under NIR irradiation, the structure generates electron/hole pairs, which are separated at the Ti₃C₂/Bi₂S₃ interface. The separated electrons then move to the MXene part of the structure and form reactive oxygen species (ROS). The manuscript reports that these ROS are able to kill bacteria. It describes a study of the antibacterial activity of the structure in-vitro and in-vivo, and reports improved healing properties of the structure when used as a wound dress. Characterization techniques were used to evaluate various properties of the structure such as structural properties, heat generation capability, ROS generation rate, antibacterial properties, and wound healing rates.

High antibacterial activity of Ti₃C₂T_x MXene has already been reported by many (e.g., 10.1021/acsnano.6b00181 & 10.1021/acssuschemeng.8b03823). These two and 7 other references listed at the end of this review are relevant, but none are cited in this manuscript. Pristine MXene has excellent antibacterial activity even in dark environment, but this manuscript does not acknowledge this and focuses only on the hybrid structure. This manuscript needs to discuss the reported antibacterial activity studies of MXenes, which will allow for determining the level of the contribution of ROS to the observed antibacterial activity. It needs to put its results into perspective, especially with respect to the antibacterial activity of other MXene-based materials (how similar or different are the antibacterial activity of this MXene/Bi₂S₃ structure?).

Apart from the major issue stated in the previous paragraph, there are several other major issues that need to be addressed thoroughly before further consideration of this manuscript.

1. "In summary, none of the as-prepared materials showed antibacterial properties independently". This statement is questionable, as there are several papers that have already reported antibacterial properties of Ti₃C₂T_x MXene (e.g., 10.1021/acsnano.6b00181& and 10.1021/acssuschemeng.8b03823). How is just 2-5-10 wt% of MXene in Bi₂S₃/MXene able to demonstrate such high antibacterial activity?
2. Figure 3 of Rasool et al. shows that the E. coli viability decreases with increased MXene concentration, but this trend is not observed in this manuscript. In the current manuscript, MB-5 has the highest antibacterial property, but MB-2 and MB-7 (with lower and higher amount of MXene, respectively) show less antibacterial activity. In fact, the antibacterial trend is not dose-dependent in this research. On the other hand, many previous studies, the antibacterial activity of the MXene has been reported to be dose dependent. What is the reason of this inconsistency?
3. Pandey et al. showed that silver-grafted nanoparticles have excellent antibacterial activity. Their nanoparticles can kill most (99%) of E. Coli bacteria. What are the advantages of Ti₃C₂/Bi₂S₃ over Ti₃C₂/Ag? A comparison between these two results would be helpful.
4. Mayerberger et al. reported antibacterial activity of a MXene against both E. coli and S. aureus (95 % and 62 % reduction in colony forming units, respectively). This means that near 30% difference in bacteria killing efficiency exist depending on bacteria type. The lower antibacterial activity against S. aureus has been attributed to the thicker peptidoglycan cell wall of the Gram-positive bacteria, which may provide protection against the antibacterial effects of the Ti₃C₂. This manuscript evaluates the antibacterial properties of Ti₃C₂/Bi₂S₃ against the same bacteria. However, the manuscript reports that Ti₃C₂/Bi₂S₃ kills 99.86% of S. aureus and 99.92% of E. coli. These two values are very close to each other. Considering the previous report on the pristine MXene, which showed 30% lower antibacterial activity against S. aureus, why does Ti₃C₂/Bi₂S₃ show a similar antibacterial activity against both E. Coli and S. Aureus?
5. There are two main mechanisms for antibacterial properties of 2D nanoparticles [physical damage of bacteria's membrane by sharp edges of the nanoparticles and the generation of

reactive oxygen species (ROS)]. This manuscript mainly focuses on improved generation of ROS. However, it does not discuss how the ability of a MXene to damage bacteria membranes physically, by its sharp edges, changes after growing Bi₂S₃ rods on MXene's surface.

6. How does the length of Bi₂S₃ rods affect the rate of electron production? In solar cells, there is a concept called exciton diffusion. Exciton is the combination of an electron and a hole. It is a big challenge in bulk-heterojunction solar cells to successfully split an electron from a hole. In these solar cells, usually made from a blend of electron donor and electron acceptor materials, the phase separation length scale of the materials should be lower than the diffusion length of exciton to give the chance of hole/electron separation before returning of the excited exciton to the ground state. Here, it seems there is a similar scenario. The electron/hole pair should be split, and electron should reach MXene before a recombination happens. This means the length of the Bi₂S₃ rods that are grown on the surface of MXene (Figure S1) should be controlled accurately. The work needs to study the effect of the Bi₂S₃ rod length on the rate of the electron generation.

7. The manuscript mentions that "The interface space only was 2.34 Å, confirming close contact." This space is schematically shown in Figure 1-C. However, this space is not observable in TEM image 1-B. The TEM figure shows a close contact between Ti₃C₂ and Bi₂S₃ phases without any space between them. This issue needs to be clarified. In addition, according to the mechanism mentioned on p.7 of the manuscript, the generated electrons in Bi₂S₃ move to a Ti₃C₂ phase. Does the 2.34 Å gap between the two phases hinder the electrons jump from the Bi₂S₃ phase to the MXene phase?

8. The manuscript states that "The corresponding photothermal conversion efficiency (η) of MB-5 reached 35.43% (Supplementary Fig. 12)". In addition, Figure 2-e shows that pristine Ti₃C₂ has lower photothermal conversion than MB-5, implying that the photothermal conversion efficiency of MXene is lower than 35.43%. This statement contradicts the near 100% light-to-heat conversion reported in Ref. (Li et al., "MXene Ti₃C₂: an effective 2D light-to-heat conversion material." ACS nano 11.4 (2017): 3752-3759).

9. The manuscript states that the first step of the growth of Bi₂S₃ is the adsorption of Bi³⁺ on the surface of MXene. The surface charge of MXene strongly depends on pH. So, the manuscript should clearly mention that at which pH this electrostatic attraction happens. The manuscript also needs to report the surface charge of Bi(NO₃)₃·5H₂O.

10. The reaction forming Bi₂S₃ should be provided. Was any covalent bond formed during the thermal treatment?

11. How were hydrodynamic sizes determined? "The diameter of Ti₃C₂T_x/Bi₂S₃ gradually decreased with the increase of Ti₃C₂T_x content", Why? As shown in TEM images, MXene nanosheets are larger than Bi₂S₃ nanorods, Also, as the method used for preparing MXene produces micron sized nanosheets, this statement seems to be incorrect.

12. As the MXene possesses O and F surface terminating groups, EDS mapping depicted in Figure 1 should also show these elements (Figure 1). The provided XPS verified the presence of O atoms.

13. Ti 2s peak in the XPS survey spectrum of the MB-5 sample is not visible, and the peak was not specified correctly.

14. Figure S8b: there should not be any O=C-O bonds in the structure of the MXene and MB-5.

15. Figure S19: It is hard to see the dead bacteria (red dots).

16. The role of physical contact (10.1021/acsnano.6b00181 & 10.1021/acssuschemeng.8b03823) should be differentiated from the factor contributing to the antibacterial activity of the prepared nanocomposite.

17. As antibacterial agents, many materials such as 2D nanomaterials of black phosphorous, MoS₂, graphene oxide, carbon nitride, and Ag-based materials have been developed. These

materials have exhibited excellent antibacterial properties. In this work, authors chose Ti₃C₂T_x MXene because of its photo-induced charge carriers. What are the advantage of Ti₃C₂T_x MXene compared with the mentioned materials?

18. "No obvious diffraction peaks of Ti₃C₂T_x were detected, which could be due to the low content and even distribution of Ti₃C₂T_x". This statement is questionable, as 10% of MXene is not a low content for collecting XRD spectra. Also, the distribution cannot affect the XRD patterns.

19. Figure S4: "The clubbed structure of Bi₂S₃ almost completely covered the ultrathin Ti₃C₂T_x nanosheets", I do not see any differences between the SEM images of Bi₂S₃ and MB-X.

20. To confirm the presence of the MXene in the structure of the nanocomposite, Raman spectroscopy is suggested. This technique is able to detect the MXene even with very low concentration of 0.05 wt% (see 10.1021/acsami.9b19960).

21. As the mass ratio of MXene is less than that of Bi₂S₃, the name of the nanocomposite should be changed to Bi₂S₃/MXene.

22. In several places in this manuscript, "Ti₂C₃T_x" should have been "Ti₃C₂T_x".

23. The sentence "Based on the results, these results can be obtained as follows" needs to be rephrased.

References

- [1] Rasool, Kashif, et al. "Antibacterial activity of Ti₃C₂T_x MXene." ACS nano 10.3 (2016): 3674-3684.
- [2] Arabi Shamsabadi, Ahmad, et al. "Antimicrobial Mode-of-Action of Colloidal Ti₃C₂T_x MXene Nanosheets." ACS Sustainable Chemistry & Engineering 6.12 (2018): 16586-16596.
- [3] Jastrzębska, A. M., et al. "In vitro studies on cytotoxicity of delaminated Ti₃C₂ MXene." Journal of hazardous materials 339 (2017): 1-8.
- [4] Lim, Gim Pao, et al. "Synthesis, characterization and antifungal property of Ti₃C₂T_x MXene nanosheets." Ceramics International (2020).
- [5] Pandey, Ravi P., et al. "Ultrahigh-flux and fouling-resistant membranes based on layered silver/MXene (Ti₃C₂T_x) nanosheets." Journal of Materials Chemistry A 6.8 (2018): 3522-3533.
- [6] Nie, Yan, et al. "MXene-Hybridized Silane Films for Metal Anticorrosion and Antibacterial Applications." Applied Surface Science (2020): 146915.
- [7] Mayerberger, Elisa A., et al. "Antibacterial properties of electrospun Ti₃C₂T_x (MXene)/chitosan nanofibers." RSC advances 8.62 (2018): 35386-35394.
- [8] Jastrzębska, Agnieszka Maria, et al. "The atomic structure of Ti₂C and Ti₃C₂ MXenes is responsible for their antibacterial activity toward E. coli bacteria." Journal of Materials Engineering and Performance 28.3 (2019): 1272-1277.
- [9] Rozmysłowska-Wojciechowska, Anita, et al. "Engineering of 2D Ti₃C₂ MXene Surface Charge and Its Influence on Biological Properties." Materials 13.10 (2020): 2347

Reviewer #2:

In this manuscript, the authors designed an interfacial Schottky junction of Ti₃C₂T_x/Bi₂S₃ with a built-in electric field resulting from the contact potential difference between Ti₃C₂T_x and Bi₂S₃. The self-driven charge transfer across the interface increases the local electron density on Ti₃C₂T_x and boosts the charge transfer and separation. The Ti₃C₂T_x/Bi₂S₃ can kill 99.86% of *Staphylococcus aureus* and 99.92% of *Escherichia coli* with the assistance of hyperthermia within 10 min. The authors have done a lot of work in this manuscript, however, some problems still exist in this work and some formats need to be further clarified.

1. It is critically necessary to compare the bacterial-killing performance of previous reported photo-thermal systems with present Ti₃C₂T_x/Bi₂S₃ photocatalysts, the advantages of Bi₂S₃ over other semiconductors should be described. The biocompatibility of the Ti₃C₂T_x/Bi₂S₃ should be

investigated.

2. The size of Bi₂S₃ was about 200 nm, does the size of Bi₂S₃ affect the performance?
3. The authors stated that the synergistic impacts of the ROS and photothermal effect killed the bacteria rapidly and effectively. The author needs to study which free radicals play a role in this system. Which is more important, ROS or photothermal effect?
4. The antibacterial properties of the Ti₃C₂T_x/Bi₂S₃ was investigated under the irradiation of the 808 nm ? What about its performance under the irradiation with other wavelength?
5. In-situ XPS should be done to evidence the built-in electric field.

Reviewer #3 (Remarks to the Author):

The manuscript entitled "Interfacial engineering of Ti₃C₂T_x MXene/Bi₂S₃ based on work function for rapid photo-excited bacteria-killing" (NCOMMS-20-22666-T) designed an interfacial Schottky junction of Ti₃C₂T_x/Bi₂S₃ with a built-in electric field. The prepared Ti₃C₂T_x/Bi₂S₃ exhibited an excellent antibacterial performance towards Staphylococcus aureus and Escherichia coli under 808 nm near-infrared light (NIR) radiation. The manuscript is well structured and the mechanisms of antibacterial performance are provided in detail. This study is of great importance and it promotes the development of the eradication of bacterial infection. Overall, the experiments have been conducted adequately. I believe that the paper is worth to be published in Nature Communications after revision based on the following:

1. - Page 5, "The interface space only was 2.34 Å, confirming close contact." Please add related references to support this expression.
2. Page 10, the author wrote that "Moreover, the homogeneous formation of Bi₂S₃ on the surface of Ti₃C₂T_x is beneficial for rapid and uniform heating." How to understand and verify the homogeneous formation?
3. How about the stability of the prepared Ti₃C₂T_x/Bi₂S₃?
4. Recent important reviews and researches can be added to better summarize the background (Introduction). Related references are listed below:
[1] Chem. Eng. J. 2019, 370, 1087-1100.
[2] Coordin. Chem. Rev. 385 (2019) 44-80.
[3] Chem. Eng. J. 2020, 382, 122840
[4] Appl. Catal. B Environ. 2020, 267, 118651.

Author's Response to Reviewer 1#

Original Comment: This manuscript describes a study on the synthesis and antibacterial activity of a Ti_3C_2 MXene/ Bi_2S_3 hybrid structure. Under NIR irradiation, the structure generates electron/hole pairs, which are separated at the $\text{Ti}_3\text{C}_2/\text{Bi}_2\text{S}_3$ interface. The separated electrons then move to the MXene part of the structure and form reactive oxygen species (ROS). The manuscript reports that these ROS are able to kill bacteria. It describes a study of the antibacterial activity of the structure *in-vitro* and *in-vivo*, and reports improved healing properties of the structure when used as a wound dress. Characterization techniques were used to evaluate various properties of the structure such as structural properties, heat generation capability, ROS generation rate, antibacterial properties, and wound healing rates.

Reply: Thank you for your valuable and professional comments on our work. We express our sincere thanks to the reviewer.

High antibacterial activity of $\text{Ti}_3\text{C}_2\text{T}_x$ MXene has already been reported by many (e.g., 10.1021/acsnano.6b00181 & 10.1021/acssuschemeng.8b03823). These two and 7 other references listed at the end of this review are relevant, but none are cited in this manuscript.

Reply: Thank you for your professional suggestion. These valuable references offer us a better understanding for the antibacterial property of $\text{Ti}_3\text{C}_2\text{T}_x$ MXene. Thus, according to your suggestion, these literatures have been cited in the revised manuscript as follows:

In **Page 3**, we have supplemented a statement that " $\text{Ti}_3\text{C}_2\text{T}_x$ MXene is an eco-friendly material with an excellent advantage due to its photo-induced charge carriers¹⁰⁻¹²."

In **Page 3**, we have supplemented these statements: "Although $\text{Ti}_3\text{C}_2\text{T}_x$ MXene has exhibited excellent antibacterial performance at least 4 h and good biocompatibility at low concentration in the dark under continuous shaking condition¹⁶⁻²², few studies

reported its antibacterial activity under NIR light."

In **Page 12**, we have supplemented these statement: "Although the recent silver/MXene nanosheets reported by Pandey et al. exhibited high antibacterial efficacy of more than 99% of *E. coli*¹⁸, as compared with current study (**Supplementary Table 2**), it was achieved at a very low bacterial concentration of 2×10^3 with much longer time of 24 h while in this work, $\text{Bi}_2\text{S}_3/\text{Ti}_3\text{C}_2\text{T}_x$ only took 10 min to kill 99.86% of *S. aureus* and 99.92% of *E. coli* with concentration of 10^7 under 808 nm NIR light irradiation."

11. Jastrzebska, A.M. et al. In vitro studies on cytotoxicity of delaminated Ti_3C_2 MXene. *J Hazard. Mater.* **339**, 1-8 (2017).
12. Rozmyslowska-Wojciechowska, A. et al. Engineering of 2D Ti_3C_2 MXene surface charge and its influence on biological properties. *Mater.* **13**, 2347 (2020).
16. Rasool, K. et al. Antibacterial activity of $\text{Ti}_3\text{C}_2\text{T}_x$ MXene. *ACS Nano* **10**, 3674-3684 (2016).
17. Arabi Shamsabadi, A., Sharifian Gh, M., Anasori, B. & Soroush, M. Antimicrobial mode-of-action of colloidal $\text{Ti}_3\text{C}_2\text{T}_x$ MXene nanosheets. *ACS Sustainable Chem. Eng.* **6**, 16586-16596 (2018).
18. Pandey, R.P. et al. Ultrahigh-flux and fouling-resistant membranes based on layered silver/MXene ($\text{Ti}_3\text{C}_2\text{T}_x$) nanosheets. *J. Mater. Chem. A* **6**, 3522-3533 (2018).
19. Mayerberger, E.A., Street, R.M., McDaniel, R.M., Barsoum, M.W. & Schauer, C.L. Antibacterial properties of electrospun $\text{Ti}_3\text{C}_2\text{T}_z$ (MXene)/chitosan nanofibers. *RSC Adv.* **8**, 35386-35394 (2018).
20. Lim, G.P. et al. Synthesis, characterization and antifungal property of $\text{Ti}_3\text{C}_2\text{T}_x$ MXene nanosheets. *Ceram. Int.* **46**, 20306-20312 (2020).
21. Nie, Y. et al. MXene-hybridized silane films for metal anticorrosion and antibacterial applications. *Appl. Surf. Sci.* **527**, (2020).
22. Jastrzebska, A.M. et al. The Atomic structure of Ti_2C and Ti_3C_2 MXenes is responsible for their antibacterial activity toward *E. coli* bacteria. *J. Mater. Eng. Perform.* **28**, 1272-1277 (2019).

Pristine MXene has excellent antibacterial activity even in dark environment, but this manuscript does not acknowledge this and focuses only on the hybrid structure.

Reply: Thank you for your professional suggestion. Just like what you said, pristine MXene does have excellent antibacterial activity even in dark environment. However, the antibacterial activity of MXene in reported literatures needs specific condition. The specific antibacterial conditions can be seen in Table R1 that we summarized from reported literatures. These references include all the literatures you mentioned. As shown in **Table R1**, the excellent antibacterial performance (the maximum antibacterial rate of 97.7% for *E.coli* and 97.04% of *B. subtilis*) is required **for 4 h even 48 h under continuous shaking**. Under these conditions, especially continuous shaking, the edge of nanosheets can cut or insert the cell membranes and vigorously extract large amounts of phospholipids from the membranes, resulting in the death of bacteria. This process can be divided into three stages. Firstly, the bacteria can tolerate nanosheets for a short period of time. And then, the cell membranes are partially damaged, with some displaying a lower surface phospholipid density but no obvious cuts. Finally, cells lose their cellular integrity; their membranes are severely damaged, and some are even missing their cytoplasm entirely. This process has been evidenced by Zhou R. et al (*Nat. Nanotechnol.* **2013**, **8**, 594-601). Indeed, this process needs a certain period of time. Additionally, the continuous shaking can accelerate the interaction between nanosheet and bacteria. The nanosheet may act like ‘blades’, which can insert and cut bacteria cell membranes. The increased shear force under higher speed can rapidly damage the integrality of cell membranes (*Sci. Adv.* **2020**, **6**, 9593; *Nat. Commun.* **2020**, **11**, 1626). **The most important is that, these antibacterial conditions are not suitable for wound healing, disinfection or sterilization in vivo and so on because the physical force and continuous shaking further damage the wound tissues.**

In contrast, in our work, the antibacterial performance was measured under static conditions with 808 nm NIR light irradiation only for 10 min. The samples only exhibited negligible effects on bacterial survival in the dark for *E. coli* and *S. aureus*, which can be evidenced by antibacterial efficiency in the dark in **Supplementary Fig. 21**. Such short time and antibacterial condition is not enough to physical damage. Thus, we focus on the antibacterial performance based on photocatalytic and photothermal therapy of the heterostructure. **The synergistic role of**

ROS and photothermal effect killed the bacteria rapidly and effectively without shaking, which is more suitable for practical application in clinic.

Meanwhile, following the suggestion of the reviewer, in the Introduction for our revised paper, we have supplemented the antibacterial activity of MXene under continuous shaking in the dark part (**Page 3**): "Although $Ti_3C_2T_x$ MXene has exhibited excellent antibacterial performance at least 4 h and good biocompatibility at low concentration in the dark under continuous shaking condition¹⁶⁻²², few studies reported the antibacterial activity under NIR light."

Table R1 The specific antibacterial conditions

Materials	Size	Concentration	Antibacterial conditions	Bacterials and antibacterial efficiency		Antibacterial mechanism	Ref.
				Gram-negative bacteria	Gram-positive bacteria		
$Ti_3C_2T_x$	-	200 $\mu\text{g/mL}$	continuous shaking at 150 rpm, constant temperature of 35°C for 4 h.	E. coli , 97.70 \pm 2.87%	B. subtilis , 97.04 \pm 2.91%	Trap or wap bacrerial, Physical damage	1
$Ti_3C_2T_x$	0.09 μm	100 $\mu\text{g/mL}$	37 °C at 150 rpm for 8 h	E. coli , >90%	B. subtilis , >90%	Physical damage	2
$Ti_3C_2T_x$ / chitosan nanofibers	298 \pm 76 nm	1.0 \pm 0.1 mg/mL	a shaking incubator at 37 C for 4 h	E. coli , over 95%	S. aureus , 62%	charged NH3 group of chitosan; Physical damage	3
Cu_2O /MXene	79 nm	40 $\mu\text{g/mL}$	37 °C at a rotating speed of 100 rpm for 20 h	P. aeruginosa 95.59%	S. aureus 97.04%	Copper(I) and copper(II) ions release; ROS; Physical damage	4
$Ti_3C_2T_x$ membranes	-	1.5 cm^2	110 rpm for up to 24 h	E. coli 67%	B. subtilis 73%	Physical damage	5
silver/MXene membranes	-	9.60 cm^2	24 h at 36 °C	E. coli more than 99%	-	TiO_2 formation; Ag nanoparticles	6
Ti_2C and Ti_3C_2 MXenes	multilayer	-	48 h at 37 °C	E. coli Ti_3C_2 , inhibition zone; Ti_2C , no	-	-	7

				antibacterial activity			
Ti ₃ C ₂ T _x MXene	-	-	11 days at 25 °C	trichoderma reesei		Physical damaging effects	8
MXene-hybridized silane films	-	-	37 °C for 24 h	E. coli	-	Oxidative stress mechanism	9

1. Rasool, K. et al. Antibacterial Activity of Ti₃C₂T_x MXene. *ACS Nano* **10**, 3674-3684 (2016).
2. Arabi Shamsabadi, A., Sharifian Gh, M., Anasori, B. & Soroush, M. Antimicrobial Mode-of-Action of Colloidal Ti₃C₂T_x MXene Nanosheets. *ACS Sustainable Chem. Eng.* **6**, 16586-16596 (2018).
3. Mayerberger, E.A., Street, R.M., McDaniel, R.M., Barsoum, M.W. & Schauer, C.L. Antibacterial properties of electrospun Ti₃C₂T_z (MXene)/chitosan nanofibers. *RSC Adv.* **8**, 35386-35394 (2018).
4. Wang, W. et al. A photo catalyst of cuprous oxide anchored MXene nanosheet for dramatic enhancement of synergistic antibacterial ability. *Chem. Eng. J.* **386**, (2020).
5. Rasool, K. et al. Efficient Antibacterial Membrane based on Two-Dimensional Ti₃C₂T_x (MXene) Nanosheets. *Sci. Rep.* **7**, 1598 (2017).
6. Pandey, R.P. et al. Ultrahigh-flux and fouling-resistant membranes based on layered silver/MXene (Ti₃C₂T_x) nanosheets. *J. Mater. Chem. A* **6**, 3522-3533 (2018).
7. Maria, J.A. et al. The atomic structure of Ti₂C and Ti₃C₂ MXenes is responsible for their antibacterial activity toward *E. coli* bacteria. *J. Mater. Eng. Perform.* **28**, (2019): 1272-1277.
8. Lim, G.P. et al. Synthesis, characterization and antifungal property of Ti₃C₂T_x MXene nanosheets. *Ceram. Int.* **46**, 20306-20312 (2020).
9. Nie, Y. et al. MXene-hybridized silane films for metal anticorrosion and antibacterial applications. *Appl. Surf. Sci.* **527**, (2020).

Supplementary Fig. 21 | Spread plate results and corresponding strain counts. a, b, Spread plate results (a) and corresponding strain counts (b) of *S. aureus*. **c, d,** Spread plate results (c) and corresponding strain counts (d) of *E. coli* treated with $Ti_3C_2T_x$, Bi_2S_3 , $Bi_2S_3/Ti_3C_2T_x-2$, $Bi_2S_3/Ti_3C_2T_x-5$, $Bi_2S_3/Ti_3C_2T_x-7$, and $Bi_2S_3/Ti_3C_2T_x-10$ under 808 nm irradiation for 10 min after 24 h co-culturing. Data are presented as mean \pm standard deviations from a representative experiment ($n = 3$ independent samples). Grey circles indicate the group of Ctrl, brown circles indicate the group of $Ti_3C_2T_x$, pink circles indicate the group of Bi_2S_3 , orange circles indicate the group of $Bi_2S_3/Ti_3C_2T_x-2$, green circles indicate the group of $Bi_2S_3/Ti_3C_2T_x-5$, blue circles indicate the group $Bi_2S_3/Ti_3C_2T_x-7$, purple circles indicate the group $Bi_2S_3/Ti_3C_2T_x-10$.

This manuscript needs to discuss the reported antibacterial activity studies of MXenes, which will allow for determining the level of the contribution of ROS to the observed antibacterial activity.

Reply: Thank you very much for the professional suggestion. Indeed, it is necessary to evaluate antibacterial activity of MXene. Thus, MXene is selected as a positive control group to measure antibacterial performance in vitro by spread plate method, bacterial live/dead fluorescence staining, and SEM antibacterial images (Supplementary Figs 21-23). There is no visual difference between MXene group and Control group.

Supplementary Fig. 21 | Spread plate results and corresponding strain counts. **a, b,** Spread plate results (**a**) and corresponding **strain counts** (**b**) of *S. aureus*. **c, d,** Spread plate results (**c**) and corresponding **strain counts** (**d**) of *E. coli* treated with $\text{Ti}_3\text{C}_2\text{T}_x$, Bi_2S_3 , $\text{Bi}_2\text{S}_3/\text{Ti}_3\text{C}_2\text{T}_x-2$, $\text{Bi}_2\text{S}_3/\text{Ti}_3\text{C}_2\text{T}_x-5$, $\text{Bi}_2\text{S}_3/\text{Ti}_3\text{C}_2\text{T}_x-7$, and $\text{Bi}_2\text{S}_3/\text{Ti}_3\text{C}_2\text{T}_x-10$ under 808 nm irradiation for 10 min after 24 h co-culturing. Data are presented as mean \pm standard deviations from a representative experiment ($n = 3$ independent samples). Grey circles indicate the group of Ctrl, brown circles indicate the group of $\text{Ti}_3\text{C}_2\text{T}_x$, pink circles indicate the group of Bi_2S_3 , orange circles indicate the group of $\text{Bi}_2\text{S}_3/\text{Ti}_3\text{C}_2\text{T}_x-2$, green circles indicate the group of $\text{Bi}_2\text{S}_3/\text{Ti}_3\text{C}_2\text{T}_x-5$, blue circles indicate the group $\text{Bi}_2\text{S}_3/\text{Ti}_3\text{C}_2\text{T}_x-7$, and purple circles indicate the group $\text{Bi}_2\text{S}_3/\text{Ti}_3\text{C}_2\text{T}_x-10$.

Supplementary Fig. 22 | Bacterial live/dead fluorescence staining of *S. aureus*. Bacterial live/dead fluorescence staining for $\text{Ti}_3\text{C}_2\text{T}_x$, Bi_2S_3 , $\text{Bi}_2\text{S}_3/\text{Ti}_3\text{C}_2\text{T}_x-2$, $\text{Bi}_2\text{S}_3/\text{Ti}_3\text{C}_2\text{T}_x-5$, $\text{Bi}_2\text{S}_3/\text{Ti}_3\text{C}_2\text{T}_x-7$, and $\text{Bi}_2\text{S}_3/\text{Ti}_3\text{C}_2\text{T}_x-10$ *in vitro* (green fluorescence represents live bacteria, red fluorescence represents dead bacteria). The scale bar is 100

Supplementary Fig. 23 | The SEM antibacterial images. a, b, SEM antibacterial images for *S. aureus* (a) and *E. coli* (b) incubated with different samples of $Ti_3C_2T_x$, Bi_2S_3 , $Bi_2S_3/Ti_3C_2T_x-2$, $Bi_2S_3/Ti_3C_2T_x-5$, $Bi_2S_3/Ti_3C_2T_x-7$, and $Bi_2S_3/Ti_3C_2T_x-10$. (The scale bar is 1 μm).

It needs to put its results into perspective, especially with respect to the antibacterial activity of other MXene-based materials (how similar or different are the antibacterial activity of this MXene/ Bi_2S_3 structure?).

Reply: Thank you very much for the professional suggestion. According to your suggestion, V_2CT_x and Nb_2CT_x were chosen as composite materials due to their good biocompatibility. The antibacterial activity of Bi_2S_3/V_2CT_x-5 and Bi_2S_3/Nb_2CT_x-5 was assessed by the same spread plate method. Firstly, Bi_2S_3/V_2CT_x-5 and Bi_2S_3/Nb_2CT_x-5 were prepared by the same hydrothermal process as for $Bi_2S_3/Ti_3C_2T_x-5$. As it can be seen in **Supplementary Fig. 29a** and **29b**, the obvious nanorods structure can be seen. And the length of Bi_2S_3/V_2CT_x-5 and Bi_2S_3/Nb_2CT_x-5 is slightly longer than $Bi_2S_3/Ti_3C_2T_x-5$. The antibacterial performance under 808 nm light irradiation was exhibited in **Supplementary Fig. 29c**. The antibacterial efficiency of Bi_2S_3/V_2CT_x-5 and Bi_2S_3/Nb_2CT_x-5 can reach to $88.23 \pm 4.39\%$, $93.03 \pm 2.43\%$, respectively. Thus, Bi_2S_3/V_2CT_x-5 and Bi_2S_3/Nb_2CT_x-5 also exhibited excellent antibacterial performance.

Besides, we supplemented these data in our revised manuscript.

In **Page 12**, we have added the statement: "Thus, the antibacterial activity of other

MXene-based materials was also assessed. As shown in Supplementary Fig. 29, the similar antibacterial activity was observed Bi₂S₃/V₂CT_x-5 and Bi₂S₃/Nb₂CT_x-5. This exciting finding definitely exhibited an enormous potential in coupling with Bi₂S₃ with a wide variety of MXene as effective antibacterial materials."

In **Supplementary Information, Page S40**, we have added these descriptions: "To verify that the Bi₂S₃ can act as a versatile antibacterial material on different MXene, V₂CT_x and Nb₂CT_x were chosen as composite materials due to their good biocompatibility^{24, 25}. The antibacterial activity of Bi₂S₃/V₂CT_x-5 and Bi₂S₃/Nb₂CT_x-5 was assessed by the same spread plate method. As it can be seen in Supplementary Fig. 29a and 29b, the obvious nanorods structure can be seen. And the length of Bi₂S₃/V₂CT_x-5 and Bi₂S₃/Nb₂CT_x-5 is slightly longer than Bi₂S₃/Ti₃C₂T_x-5. The antibacterial performance under 808 nm light irradiation was exhibited in Supplementary Fig. 29c. The antibacterial efficiency of Bi₂S₃/V₂CT_x-5 and Bi₂S₃/Nb₂CT_x-5 can reach to 88.23 ± 4.39%, 93.03 ± 2.43%, respectively. Thus, Bi₂S₃/V₂CT_x-5 and Bi₂S₃/Nb₂CT_x-5 also exhibited excellent antibacterial performance."

Supplementary Fig. 29 | The characterization and antibacterial properties of Bi₂S₃/V₂CT_x-5 and Bi₂S₃/Nb₂CT_x-5. a, SEM images and corresponding EDS analysis of Bi₂S₃/V₂CT_x-5. b, SEM images and corresponding EDS analysis of Bi₂S₃/Nb₂CT_x-5. c, The spread plate results and corresponding strain counts of *S. aureus* under 808 nm light irradiation (0.7 W cm⁻²).

24. Zada, S. et al. Algae extraction controllable delamination of Vanadium Carbide nanosheets with enhanced near - infrared photothermal performance. *Angew.*

Chem. **132**, 6663-6668 (2020).

25. Yin, H. et al. Nanomedicine-enabled photonic thermogaseous cancer therapy. *Adv. Sci.* **7** (2019).

Apart from the major issue stated in the previous paragraph, there are several other major issues that need to be addressed thoroughly before further consideration of this manuscript.

Reply: Thank you very much for the professional suggestion. Those comments are very helpful for us to revise and improve our paper. According to these comments, we have made extensive and careful revisions one by one and sincerely hope to get your reconsideration.

Comment 1: "In summary, none of the as-prepared materials showed antibacterial properties independently". This statement is questionable, as there are several papers that have already reported antibacterial properties of $Ti_3C_2T_x$ MXene (e.g., 10.1021/acsnano.6b00181 & and 10.1021/acssuschemeng.8b03823). How is just 2-5-10 wt% of MXene in Bi_2S_3 /MXene able to demonstrate such high antibacterial activity?

Reply: Thank you very much for your professional suggestion. We are sorry for our negligence for inaccurate statement. Indeed, these materials did not show antibacterial properties under certain conditions. Thus, we revised the corresponding description in **Page 11:** "In summary, none of the as-prepared materials for ten minutes under static conditions showed antibacterial properties."

As the aforementioned response, we agree with this point that $Ti_3C_2T_x$ MXene has antibacterial properties. Pristine MXene nanosheet showed good antibacterial performance under continuous shaking at least 4 h. These conditions can provide more chance to use the sharp edges to damage bacteria, resulting in the death of bacteria. This antibacterial mechanism needs to provide longer time for the material to interact with the bacteria (*Nat. Nanotechnol.* **2013**, **8**, 594-601). In present work, the interaction only has ten minutes under static conditions. It is difficult for MXene to cut bacteria using the edges in such a short time. Thus, it is not contradictory to different antibacterial

performance. The result also is demonstrated by antibacterial performance in the dark (Supplementary Fig. 21-23). All samples showed almost no antibacterial properties for ten minutes under static conditions.

Supplementary Fig. 21 | Spread plate results and corresponding strain counts. a, b, Spread plate results (a) and corresponding strain counts (b) of *S. aureus*. **c, d,** Spread plate results (c) and corresponding strain counts (d) of *E. coli* treated with $Ti_3C_2T_x$, Bi_2S_3 , $Bi_2S_3/Ti_3C_2T_x-2$, $Bi_2S_3/Ti_3C_2T_x-5$, $Bi_2S_3/Ti_3C_2T_x-7$, and $Bi_2S_3/Ti_3C_2T_x-10$ under 808 nm irradiation for 10 min after 24 h co-culturing. Data are presented as mean \pm standard deviations from a representative experiment ($n = 3$ independent samples). Grey circles indicate the group of Ctrl, brown circles indicate the group of $Ti_3C_2T_x$, pink circles indicate the group of Bi_2S_3 , orange circles indicate the group of $Bi_2S_3/Ti_3C_2T_x-2$, green circles indicate the group of $Bi_2S_3/Ti_3C_2T_x-5$, blue circles indicate the group $Bi_2S_3/Ti_3C_2T_x-7$, and purple circles indicate the group $Bi_2S_3/Ti_3C_2T_x-10$.

Supplementary Fig. 22 | Bacterial live/dead fluorescence staining of *S. aureus*.

Bacterial live/dead fluorescence staining for $\text{Ti}_3\text{C}_2\text{T}_x$, Bi_2S_3 , $\text{Bi}_2\text{S}_3/\text{Ti}_3\text{C}_2\text{T}_x-2$, $\text{Bi}_2\text{S}_3/\text{Ti}_3\text{C}_2\text{T}_x-5$, $\text{Bi}_2\text{S}_3/\text{Ti}_3\text{C}_2\text{T}_x-7$, and $\text{Bi}_2\text{S}_3/\text{Ti}_3\text{C}_2\text{T}_x-10$ *in vitro* (green fluorescence represents live bacteria, red fluorescence represents dead bacteria). The scale bar is 100 μm .

Supplementary Fig. 23 | The SEM antibacterial images. a, b, SEM antibacterial images for *S. aureus* (a) and *E. coli* (b) incubated with different samples of $\text{Ti}_3\text{C}_2\text{T}_x$, Bi_2S_3 , $\text{Bi}_2\text{S}_3/\text{Ti}_3\text{C}_2\text{T}_x-2$, $\text{Bi}_2\text{S}_3/\text{Ti}_3\text{C}_2\text{T}_x-5$, $\text{Bi}_2\text{S}_3/\text{Ti}_3\text{C}_2\text{T}_x-7$, and $\text{Bi}_2\text{S}_3/\text{Ti}_3\text{C}_2\text{T}_x-10$. (The scale bar is 1 μm).

To such high antibacterial activity, the manuscript has been systematically and comprehensively analyzed this reason. It can be summarized as following:

1. The heterojunction between Bi_2S_3 and $\text{Ti}_3\text{C}_2\text{T}_x$ reduced the recombination photogenerated electrons and holes (**Fig. 2c**), more electrons can participate in the generation of reactive oxygen species. Hence, the yield of reactive oxygen species can be drastically enhanced (**Fig. 2a**), resulting in more free radicals to damage bacteria.

2. The introduction of $\text{Ti}_3\text{C}_2\text{T}_x$ induced the formation of the impurity levels in the band gap of Bi_2S_3 (**Figs. 4e and 4f**), reducing the band gap of Bi_2S_3 (**Supplementary Fig. 16**). This can reduce the energy of electron jump and accelerate the generation of photogenerated electrons. More electrons can participate in the formation of reactive oxygen species.

3. The synergistic effect of the deep-level defects of Bi_2S_3 and the localized surface plasmon resonance of $\text{Ti}_3\text{C}_2\text{T}_x$ obviously improved the photothermal effects (**Fig. 2e**),

which can decrease the viability of bacteria.

In short, the synergistic effect of these factors can achieve rapid and effective antibacterial effect in 10 min under 808 nm light irradiation.

Fig. 2 | The photocatalytic and photothermal properties of Ti₃C₂T_x, Bi₂S₃, Bi₂S₃/Ti₃C₂T_x-2, Bi₂S₃/Ti₃C₂T_x-5, Bi₂S₃/Ti₃C₂T_x-7, and Bi₂S₃/Ti₃C₂T_x-10. a, ROS production with DCFH fluorescence probe (200 ppm). b, PL spectra tested with 325 nm excitation wavelength from 350 nm to 600 nm. c, Photothermal curves of different samples under 10 min light (808 nm) irradiation.

Fig. 4 | DFT calculation of the Bi₂S₃/Ti₃C₂F₂ and Bi₂S₃/Ti₃C₂O₂ structure. e, f, Electronic band gap structure of Bi₂S₃/Ti₃C₂F₂ and Bi₂S₃/Ti₃C₂O₂. Blue-grey colors denote contributions from the bulk part of the Ti₃C₂F₂ (e) and Ti₃C₂O₂ (f). The darkness of this color is determined by the degree of contribution.

Supplementary Fig. 16 | Energy gap calculated from UV-vis-NIR diffuse reflectance spectra.

Comment 2: Figure 3 of Rasool et al. shows shows that the *E. coli* viability decreases with increased MXene concentration, but this trend is not observed in this manuscript. In the current manuscript, MB-5 has the highest antibacterial property, but MB-2 and MB-7 (with lower and higher amount of MXene, respectively) show less antibacterial activity. In fact, the antibacterial trend is not dose-dependent in this research. On the other hand, many previous studies, the antibacterial activity of the MXene has been reported to be dose dependent. What is the reason of this inconsistency?

Reply: Thank you very much for your professional suggestion. As you said, the antibacterial activity of MXene under constant shaking for the dark depends on concentration. These references (*ACS Nano* 2016, 10, 3674–3684; *Sci. Rep.* 2017, 7, 1598) have reported this point. Rasool *et al.* exhibited the antibacterial activity of $\text{Ti}_3\text{C}_2\text{T}_x$ modified membranes against *E. coli* and *B. subtilis*, which can reach more than 73% against *B. subtilis* and 67% against *E. coli* under constant shaking at 110 rpm for

up to 24 h. With the increase of MXene concentration, the more chance can be provided to interact with bacteria. The sharp edges of MXene can damage more bacteria to reach excellent antibacterial activity at same time. So, the antibacterial trend depends on dose under dark condition.

In contrast, in our work, the antibacterial mechanism is not physical damage, but the synergy of both photocatalytic and photothermal effect, which plays a vital role for the death of bacteria. Such a small amount of $\text{Ti}_3\text{C}_2\text{T}_x$ for $\text{Bi}_2\text{S}_3/\text{Ti}_3\text{C}_2\text{T}_x$ is not enough to reach physical damage within 10 min, especially under static conditions. Therefore, there is no obvious trend for different samples in spite of the increase of $\text{Ti}_3\text{C}_2\text{T}_x$ in $\text{Bi}_2\text{S}_3/\text{Ti}_3\text{C}_2\text{T}_x$. In addition, the antibacterial performance of $\text{Bi}_2\text{S}_3/\text{Ti}_3\text{C}_2\text{T}_x$ also depends on concentration. As it can be seen in **Supplementary Fig. 20**, the antibacterial rare gradually increased as the concentration increases under light irradiation. The higher concentration can provide more heat and ROS to kill bacteria, thus reaching excellent antibacterial efficiency.

Supplementary Fig. 20 | Photographs of bacterial colonies and corresponding strain counts. **a, b**, Spread plate results (**a**) and corresponding strain counts (**b**) of *S. aureus*. **c, d**, Spread plate results (**c**) and corresponding strain counts (**d**) of *E. coli* treated with different concentration of $\text{Bi}_2\text{S}_3/\text{Ti}_3\text{C}_2\text{T}_x$ -5 under 808 nm irradiation for 10 min after 24 h co-culturing. Data are presented as mean \pm standard deviations from a

representative experiment ($n = 3$ independent samples). Grey circles indicate the group of Ctrl, brown circles indicate the group of 50 ppm, pink circles indicate the group of 100 ppm, and green circles indicate the group of 200 ppm.

Comment 3: Pandey et al. showed that silver-grafted nanoparticles have excellent antibacterial activity. Their nanoparticles can kill most (99%) of *E. Coli* bacteria. What are the advantages of $\text{Ti}_3\text{C}_2/\text{Bi}_2\text{S}_3$ over $\text{Ti}_3\text{C}_2/\text{Ag}$? A comparison between these two results would be helpful.

Reply: Thanks a lot for your professional comment. Following your suggestion, we compared the antibacterial system between $\text{Bi}_2\text{S}_3/\text{Ti}_3\text{C}_2\text{T}_x$ and $\text{Ti}_3\text{C}_2/\text{Ag}$. The results of this comparison are listed in **Supplementary Table 2**.

Supplementary Table 2 The comparison between $\text{Bi}_2\text{S}_3/\text{Ti}_3\text{C}_2\text{T}_x$ and $\text{Ti}_3\text{C}_2/\text{Ag}$

	$\text{Bi}_2\text{S}_3/\text{Ti}_3\text{C}_2\text{T}_x$	$\text{Ti}_3\text{C}_2/\text{Ag}$ membranes
Materials	Bi_2S_3 , 5wt% $\text{Ti}_3\text{C}_2\text{T}_x$	21% Ag, Ti_3C_2
Concentration	200 ppm	-
Condition	10 min, 808 nm irradiation	24 h at 36°C
Bacterial concentration (CFU mL^{-1})	10^7	2×10^3
Antibacterial rate	99.86% of S. aureus , 99.92% of E. coli	more than 99% E. coli
Antibacterial mechanism	The synergistic effect of Photodynamic and photothermal role	The release of Ag, contact with bacteria
Biocompatibility	Good	-

As shown in **Supplementary Table 2**, $\text{Bi}_2\text{S}_3/\text{Ti}_3\text{C}_2\text{T}_x$ **only took 10 min** to kill 99.86% of *S. aureus* and 99.92% of *E. coli* by synergy of ROS and hyperthermia under

808 nm light irradiation while keeping good cytocompatibility simultaneously, avoiding the formation of biofilm and the emergence of super bacteria. However, the reported $\text{Ti}_3\text{C}_2/\text{Ag}$ took **much more time of 24 h** to kill more than 99% of *E. coli* with a very low bacterial concentration of 2×10^3 (far lower than the one of 10^7 in our work) by released Ag¹⁸. **The most important is that Ag has strong cytotoxicity and is easy to accumulate in the body, causing toxic side effects.** Hence, $\text{Bi}_2\text{S}_3/\text{Ti}_3\text{C}_2\text{T}_x$ in our work is much better than $\text{Ti}_3\text{C}_2/\text{Ag}$ for antibacterial application in clinic.

In our revised paper, we supplemented the following description in **Page 12**:

"Although the recent silver/MXene nanosheets reported by Pandey et al. exhibited high antibacterial efficacy of more than 99% of *E. coli*¹⁸, as compared with current study (**Supplementary Table 2**), it was achieved at a very low bacterial concentration of 2×10^3 with much longer time of 24 h while in this work, $\text{Bi}_2\text{S}_3/\text{Ti}_3\text{C}_2\text{T}_x$ only took 10 min to kill 99.86% of *S. aureus* and 99.92% of *E. coli* with concentration of 10^7 under 808 nm NIR light irradiation. Furthermore, the bacteria were killed by released Ag and physical action of MXene nanosheets in the former, which exhibited potential toxicity from released Ag. It has been reported that long time contact with bacteria can also induce bacterial resistance, even for Ag⁴⁷. In contrast, $\text{Bi}_2\text{S}_3/\text{Ti}_3\text{C}_2\text{T}_x$ killed bacteria rapidly through synergy of ROS and hyperthermia under 808 nm light irradiation with good cytocompatibility simultaneously."

18. Pandey, R.P. et al. Ultrahigh-flux and fouling-resistant membranes based on layered silver/MXene ($\text{Ti}_3\text{C}_2\text{T}_x$) nanosheets. *J. Mater. Chem. A* **6**, 3522-3533 (2018).

47. Panáček, A. et al. Bacterial resistance to silver nanoparticles and how to overcome it. *Nat. Nanotechnol.* **13**, 65-71 (2018).

Comment 4: Mayerberger et al. reported antibacterial activity of a MXene against both *E. coli* and *S. aureus* (95% and 62% reduction in colony forming units, respectively). This means that near 30% difference in bacteria killing efficiency exist depending on bacteria type. The lower antibacterial activity against *S. aureus* has been attributed to the thicker peptidoglycan cell wall of the Gram-positive bacteria, which may provide

protection against the antibacterial effects of the Ti_3C_2 . This manuscript evaluates the antibacterial properties of Ti_3C_2/Bi_2S_3 against the same bacteria. However, the manuscript reports that Ti_3C_2/Bi_2S_3 kills 99.86% of *S. aureus* and 99.92% of *E. coli*. These two values are very close to each other. Considering the previous report on the pristine MXene, which showed 30% lower antibacterial activity against *S. aureus*, why does Ti_3C_2/Bi_2S_3 show a similar antibacterial activity against both *E. coli* and *S. aureus*?

Reply: Thank you very much for the professional suggestion. Indeed, the antibacterial performance of $Ti_3C_2T_z/CS$ composite nanofibers reported by Mayerberger et al. had obvious difference against *E. coli* and *S. aureus* compared with our work. The loading of $Ti_3C_2T_z$ was only 0.75 wt% of CS. The antibacterial tests were measured **in a shaking incubator** at 37 °C for 4 h. The inherent antibacterial activity (the charged NH_3 group) of CS and direct contact of $Ti_3C_2T_x$ flakes was considered to be most likely mechanisms. As you have said, the thicker peptidoglycan cell wall of *S. aureus* needs more physical damage, leading to a lower antibacterial rate.

In our manuscript, the synergy of ROS and hyperthermia are the main antibacterial mechanisms. ROS can cause photo-oxidative stress for bacteria, which is related to two candidate genes (glutathione peroxidase and Zn-dependent hydrolase of the glyoxalase II family) (*Nat. Rev. Microbiol.* 2009, 7, 856-863; *Nat. Rev. Microbiol.* 2017, 15, 385-396.). Oxidative damage to proteins can result in the production of protein peroxides. Hyperthermia induced by photothermal effect can significantly accelerate Glutathione oxidation, which breaks down the bacterial antioxidant defense system. These effects have no obvious relationship with the type of bacteria. Thus, there is similar antibacterial activity against both *E. coli* and *S. aureus*.

Comment 5: There are two main mechanisms for antibacterial properties of 2D nanoparticles [physical damage of bacteria's membrane by sharp edges of the nanoparticles and the generation of reactive oxygen species (ROS)]. This manuscript mainly focuses on improved generation of ROS. However, it does not discuss how the ability of a MXene to damage bacteria membranes physically, by its sharp edges,

changes after growing Bi₂S₃ rods on MXense's surface.

Reply: Thanks a lot for your professional comment. This is a very good suggestion. In this work, the main mechanisms for antibacterial performance are the synergistic effect of reactive oxygen species and photothermal effect. The physical damage of Bi₂S₃/Ti₃C₂T_x to bacteria's membrane was not assessed. In our revised paper, the experiment was supplemented in **Supplementary Fig. 25** and the corresponding description was added in **Page 11**: "Additionally, the samples in the dark for 12 h did not exhibit antibacterial activity (Supplementary Fig. 25)."

The experimental method was supplemented in **Page 19**: "The 10⁷ CFU mL⁻¹ each of *E. coli* and *S. aureus* in the dark were suspended in samples (200 ppm) and cultured under continuous shaking at 120 rpm and a constant temperature of 37 °C for 12 h to assess the antibacterial performance."

Based on above results, Bi₂S₃/Ti₃C₂T_x did not exhibit obvious physical damage to *E. coli* and *S. aureus*. The physical mode of the bacterial cell wall and material surface that can achieve physical bacteria-killing can be divided into nanostructure-induced stretching of the bacterial cell and cutting of the bacterial cell membrane by the action of sharp nano-edges (*Nat. Rev. Microbiol.* 2020). The structure that nanorods *in-situ* grown on the surface of nanosheet reduces the edge sharpness, thus weakening the effect of physical damage. Especially, this antibacterial mode under continuous shaking is not suitable for the application of wound healing because the continuous shaking may lead to secondary damage to wound tissues.

Supplementary Fig. 25 | The antibacterial performance in the dark. Spread plate results and corresponding strain counts of *S. aureus* and *E. coli* for 12 h in the dark. Data are shown as mean \pm standard deviations; $n = 3$ independent samples. Source data are provided as a Source Data file.

Comment 6: How does the length of Bi_2S_3 rods affect the rate of electron production? In solar cells, there is a concept called exciton diffusion. Exciton is the combination of an electron and a hole. It is a big challenge in bulk-heterojunction solar cells to successfully split an electron from a hole. In these solar cells, usually made from a blend of electron donor and electron acceptor materials, the phase separation length scale of the materials should be lower than the diffusion length of exciton to give the chance of hole/electron separation before returning of the excited exciton to the ground state. Here, it seems there is a similar scenario. The electron/hole pair should be split, and electron should reach MXene before a recombination happens. This means the length of the Bi_2S_3 rods that are grown on the surface of MXene (Figure S1) should be controlled accurately. The work needs to study the effect of the Bi_2S_3 rod length on the rate of the electron generation.

Reply: Thanks a lot for your professional comment. This is a very good suggestion. The length of Bi_2S_3 rods may have an effect on the separation efficiency of photoexcited electron-hole pairs. Thus, in our revised paper, we supplemented the antibacterial

experiments of the different length of Bi₂S₃/Ti₃C₂T_x-5 in **Supplementary Fig. 30**, and the rate of electron generation was measured. The result demonstrated that the photogenerated electrons gradually increased with the increase of Bi₂S₃ length. This may be attributed to the different incident photon-to-current conversion efficiency. The different length of TiO₂ nanowires also exhibited the similar trend (*ACS Nano*, 2012, **6**, 5060-5069).

In our revised paper, we added the statement in **Page S41 in Supplementary Information**:

"To demonstrate the effect of Bi₂S₃ rod lengths on the rate of photogenerated electrons, the different lengths of Bi₂S₃ was prepared. As shown in Supplementary Fig. 30a, the different lengths can be observed from particles to long rods (more than 1 μm). PL spectra can reflect the separation efficiency of photogenerated electrons and holes. The longer rods showed a lower photogenerated electron-hole recombination rate, indicating the efficient transfer and separation of photoexcited electron-hole. Similarly, there was an increasing tendency in photocurrent with the increase of length. PL spectra and photocurrent density showed that the length of Bi₂S₃ had an influence on the separation of electron-hole pairs. This similar result was also demonstrated by TiO₂ nanowires length²⁶."

26. Hwang, Y. J., Hahn, C., Liu, B., & Yang, P. Photoelectrochemical properties of TiO₂ nanowire arrays: a study of the dependence on length and atomic layer deposition coating. *ACS Nano* **6**, 5060-5069, (2012).

Supplementary Fig. 30 | The characterization and photocatalytic properties of Bi₂S₃/Ti₃C₂T_x-5 with different rods. a, SEM images b, PL spectra tested with 325 nm excitation wavelength from 350 nm to 600 nm. c, Photocurrent response under 808 nm NIR irradiation.

Comment 7: The manuscript mentions that "The interface space only was 2.34 Å, confirming close contact." This space is schematically shown in Figure 1-C. However, this space is not observable in TEM image 1-B. The TEM figure shows a close contact between Ti₃C₂ and Bi₂S₃ phases without any space between them. This issue needs to be clarified. In addition, according to the mechanism mentioned on p.7 of the manuscript, the generated electrons in Bi₂S₃ move to a Ti₃C₂ phase. Does the 2.34 Å gap between the two phases hinder the electrons jump from the Bi₂S₃ phase to the MXene phase?

Reply: Thanks a lot for your professional comment. This is a very good question.

The spacing between Ti₃C₂T_x and Bi₂S₃ cannot hinder the electrons jump, only has a low impact on electron transfer. Because electrons or holes only need to move less than 1 nm vertically for the charge transfer process to happen (Nat. Nanotechnol. 9, 682-686 (2014)). The space of 2.34 Å was an equilibrium state under our computational conditions. Bi₂S₃/Ti₃C₂T_x structure can form Schottky junction. With

the decreasing or increasing of interfacial distance, the Bi₂S₃/Ti₃C₂T_x structure is under compressive or tensile strain, which can tune the energy band and Schottky barrier, further affecting the energy of electrons transfer from Bi₂S₃ to Ti₃C₂T_x. These results have been evidenced by the result of graphene–blue phosphorus (*Jpn. J. Appl. Phys.* **55**, 080306 (2016)).

Actually, the interface space between Ti₃C₂T_x and Bi₂S₃ was calculated by density functional theory (DFT) in this work. The interface space was affected by energy cutoff, convergence criterion of geometry relaxation on each atom, and calculation functional (Perdew-Burke-Emzerhof (PBE), local density approximation (LDA), and van der Waals density functional 2 (vdW-DF2)). In our manuscript, DFT calculations were performed by Projector Augmented Wave (PAW) potentials and the electron exchange correlation interactions described by the generalized gradient approximation (GGA) with PBE functional. The convergence criterion of geometry relaxation was set to 0.01 eV•Å⁻¹ in force on each atom. The energy cutoff for plane wave-basis was set to 500 eV. After full relaxation, the optimized configurations of Bi₂S₃/Ti₃C₂T_x can be obtained. In this system, the interlayer spacing of Bi₂S₃/Ti₃C₂T_x (F, O, OH) was 2.62, 2.51, 1.43 Å, respectively. According to the proportion of F, O, OH from XPS results, the average of space can be calculated to be 2.34 Å. So we are sorry for our unclear statements.

We have modified the statements in **Page 5**: "A stable crystal structure was formed and the average interface space was about 2.34 Å, confirming close contact^{32, 33}."

The similar results can be seen in other references. As shown in Figure R1, the distance between graphene and MoS₂ can vary with calculation functional. Thus, the distance may not be observed in TEM images in practice.

1. *J. Phys. Chem. C* **2015**, *119*, 19928-19933

Figure R1. Binding energy curves for graphene/MoS₂ calculated using the PBE, LDA, and vdW-DF2 xc functionals.

Comment 8: The manuscript states that "The corresponding photothermal conversion efficiency (η) of MB-5 reached 35.43% (Supplementary Fig. 12)". In addition, Figure 2-e shows that pristine Ti₃C₂ has lower photothermal conversion than MB-5, implying that the photothermal conversion efficiency of MXene is lower than 35.43%. This statement contradicts the near 100% light-to-heat conversion reported in Ref. (Li et al., "MXene Ti₃C₂: an effective 2D light-to-heat conversion material." ACS nano 11.4 (2017): 3752-3759).

Reply: Thank you very much for your professional suggestion. As the reviewer said, the internal light-to-heat conversion efficiency of Ti₃C₂T_x can reach to near 100% in the reported literature. The difference was resulted from the different system. We have compared the difference of measured method and calculation. Figure R2 shows the difference clearly. Firstly, in droplet-based light absorption, Ti₃C₂T_x was hung at the tip of one-end-sealed PTFE pipet (ACS nano 11.4 (2017): 3752-3759). And parallel light source is shone right in the center of the droplet (**Figure R2a**). In contrast, in our system, 808 nm light source irradiated the container containing Ti₃C₂T_x (**Figure R2b**), inevitably causing the temperature change of container (*J. Am. Chem. Soc.* 2017, 139, 16235–16247; *J. Phys. Chem. C* 2007, 111, 3636-3641). These differences of device

lead to the difference of light-to-heat conversion efficiency. Thus, the calculated light-to-heat conversion efficiency (η) needs to subtract the heating absorption energy of container. Li *et al.* obtained the finally calculated equation as follows:

$$\eta = \frac{F(T_{max}-T_0)}{W(1-10^{-A})} \quad (1)$$

where F is the proportional coefficient that describes heat loss process, T_{max} is the maximum droplet temperature, T_0 is the surrounding temperature, W is power of the incident laser beam, A is absorbance measured by directly on the spectrophotometer.

According the equation (7) in this literature, that is

$$F = -\frac{\ln \frac{T-T_0}{T_{max}-T_0}}{t} mc \quad (2)$$

where t is time, T is the temperature of the droplet, m is the total mass of droplet, C denotes the specific heat capacity of H₂O. Thus, the final calculated equation can be inferred as follows:

$$\eta = \frac{F(T_{max}-T_0)}{W(1-10^{-A})} = \frac{-\frac{\ln \frac{T-T_0}{T_{max}-T_0}}{t} mc (T_{max}-T_0)}{W(1-10^{-A})} \quad (3)$$

However, in our system, the light-to-heat conversion efficiency (η) was calculated using equation (4)

$$\eta = \frac{hS(T_{max} - T_0) - Q_0}{W(1 - 10^{-A})} = \frac{-\frac{\ln \frac{T-T_0}{T_{max}-T_0}}{t} mc (T_{max} - T_0) - Q_0}{W(1 - 10^{-A})} \quad (4)$$

where Q_0 indicates the heating absorption energy of container

From the equation (3) and (4), the heating absorption energy of container is the main difference of light-to-heat conversion efficiency.

In addition, other MXene also was reported by this method. The light-to-heat conversion efficiency of MXenes and their nanocomposites are summarized in **Table R2**.

Figure R2. Experimental set up for light-to-heat conversion experiment. **a**, Droplet-based light-to-heat conversion experiment¹. **b**, our experimental equipment.

1. Li, R., Zhang, L., Shi, L. & Wang, P. MXene Ti_3C_2 : An effective 2d light-to-heat conversion material. *ACS Nano* **11**, 3752-3759 (2017).

Table R2. The light-to-heat conversion efficiency of MXenes and their nanocomposites.

Materials	Light source	Light-to-heat conversion efficiency	References
Core-shell $\text{Ti}_3\text{C}_2@ \text{Au-PEG}$ nanosheets	1064 nm, 0.75 W cm^{-2}	39.6%	1
$\text{Ti}_3\text{C}_2@m\text{MSNs}$	808 nm, 1.5 W cm^{-2}	23.2%	2
$\text{Ta}_4\text{C}_3\text{-SP}$ nanosheets	808 nm, 1.5 W cm^{-2}	44.7%	3
$\text{Ta}_4\text{C}_3\text{-IONP-SPs}$	808 nm, 1.5 W cm^{-2}	32.5%	4
$\text{MnO}_x/\text{Ta}_4\text{C}_3$	808 nm, 1.5 W cm^{-2}	34.9%	5
$\text{GdW}_{10}@ \text{Ti}_3\text{C}_2$	808 nm, 1.5 W cm^{-2}	21.9%	6
$\text{CTAC}@ \text{Nb}_2\text{C-MSN}$	1064 nm, 1.5 W cm^{-2}	28.6%	7
Nb_2C nanosheets	808 nm, 1.5 W cm^{-2}	36.4%	8
$\text{Ti}_3\text{C}_2\text{-DOX}$	808 nm, 1.5 W cm^{-2}	58.3%	9
$\text{MnO}_x/\text{Ti}_3\text{C}_2\text{-SP}$ composite nanosheets	808 nm, 1.0 W cm^{-2}	22.9%	10
$\text{PLGA}/\text{Ti}_3\text{C}_2$	808 nm, 1.5 W cm^{-2}	30.6%	11

1. Tang, W. et al. Multifunctional Two-dimensional core-shell MXene@Gold nanocomposites for enhanced photo-radio combined therapy in the second biological window. *ACS Nano* **13**, 284-294 (2019).
2. Li, Z. et al. Surface nanopore engineering of 2D MXenes for targeted and synergistic multitherapies of hepatocellular carcinoma. *Adv. Mater.* **30**, e1706981 (2018).
3. Lin, H., Wang, Y., Gao, S., Chen, Y. & Shi, J. Theranostic 2D Tantalum Carbide (MXene). *Adv. Mater.* **30** (2018).
4. Liu, Z. et al. 2D Superparamagnetic Tantalum Carbide composite MXenes for efficient breast-cancer theranostics. *Theranostics* **8**, 1648-1664 (2018).
5. Dai, C. et al. Two-Dimensional Tantalum Carbide (MXenes) Composite nanosheets for multiple imaging-guided photothermal tumor ablation. *ACS Nano* **11**, 12696-12712 (2017).
6. Zong, L., Wu, H., Lin, H. & Chen, Y. A polyoxometalate-functionalized two-dimensional titanium carbide composite MXene for effective cancer theranostics. *Nano Res.* **11**, 4149-4168 (2018).
7. Han, X. et al. Therapeutic mesopore construction on 2D Nb₂C MXenes for targeted and enhanced chemo-photothermal cancer therapy in NIR-II biowindow. *Theranostics* **8**, 4491-4508 (2018).
8. Lin, H., Gao, S., Dai, C., Chen, Y. & Shi, J. A Two-dimensional biodegradable Niobium Carbide (MXene) for photothermal tumor eradication in NIR-I and NIR-II biowindows. *J Am. Chem. Soc.* **139**, 16235-16247 (2017).
9. Liu, G. et al. Surface Modified Ti₃C₂ MXene Nanosheets for tumor targeting photothermal/photodynamic/chemo synergistic therapy. *ACS Appl. Mater. Interfaces* **9**, 40077-40086 (2017).
10. Dai, C. et al. Biocompatible 2D Titanium Carbide (MXenes) composite nanosheets for pH-responsive MRI-guided tumor hyperthermia. *Chem. Mater.* **29**, 8637-8652 (2017).
11. Lin, H., Wang, X., Yu, L., Chen, Y. & Shi, J. Two-dimensional ultrathin MXene ceramic nanosheets for photothermal conversion. *Nano Lett.* **17**, 384-391 (2017).

Comment 9: The manuscript states that the first step of the growth of Bi₂S₃ is the adsorption of Bi³⁺ on the surface of MXene. The surface charge of MXene strongly depends on pH. So, the manuscript should clearly mention that at which pH this electrostatic attraction happens. The manuscript also needs to report the surface charge

of $\text{Bi}(\text{NO}_3)_3 \cdot 5\text{H}_2\text{O}$.

Reply: Thank you very much for your professional suggestion. We are sorry for our negligence and we supplemented corresponding pH value of $\text{Ti}_3\text{C}_2\text{T}_x$ MXene in **Page 15 in our revised paper: "Synthesis of $\text{Bi}_2\text{S}_3/\text{Ti}_3\text{C}_2\text{T}_x$** . To synthesize the $\text{Bi}_2\text{S}_3/\text{Ti}_3\text{C}_2\text{T}_x$ composites, $\text{Ti}_3\text{C}_2\text{T}_x$ aqueous solution (pH=6.42) was sonicated by ultrasonic cell grinder." The surface charge of $\text{Bi}(\text{NO}_3)_3 \cdot 5\text{H}_2\text{O}$ has been provided in **Supplementary Fig. 5**. The value is 34.29 ± 2.13 mV.

Supplementary Fig. 5 | Zeta potential. Zeta potential of the $\text{Ti}_3\text{C}_2\text{T}_x$ suspension, $\text{Bi}(\text{NO}_3)_3 \cdot 5\text{H}_2\text{O}$, $\text{Bi}_2\text{S}_3/\text{Ti}_3\text{C}_2\text{T}_x-2$, $\text{Bi}_2\text{S}_3/\text{Ti}_3\text{C}_2\text{T}_x-5$, $\text{Bi}_2\text{S}_3/\text{Ti}_3\text{C}_2\text{T}_x-7$, and $\text{Bi}_2\text{S}_3/\text{Ti}_3\text{C}_2\text{T}_x-10$. Data are shown as mean \pm standard deviations; $n = 3$ independent samples.

Comment 10: The reaction forming Bi_2S_3 should be provided. Was any covalent bond formed during the thermal treatment?

Reply: Thank you very much for your professional suggestion. The nucleation and growth of Bi_2S_3 nanorods are based on Bi^{3+} and S^{2-} ions released slowly from the precursor solution. Thioacetamide (TAA) is used as both sulfur source and ligand, which can form complexes $[\text{Bi}(\text{TAA})_n]^{3+}$ with Bi^{3+} . During hydrothermal process, the

following reactions may occur in the solution.

We have added these statements in **Page S3 in Supplementary Information**: "During hydrothermal treatment, the nucleation and growth of Bi_2S_3 nanorods are based on Bi^{3+} and S^{2-} ions released slowly from the precursor solution. Thioacetamide (TAA) is used as both sulfur source and ligand, which can form complexes $[\text{Bi}(\text{TAA})_n]^{3+}$ with Bi^{3+} . During hydrothermal process, the following reactions may occur in the solution.

The structure of Bi_2S_3 determines the morphology of the material, which comprises a macromolecular lamellar net with trigonally coordinated Bi and S atoms (*Chem. Mater.* **2003**, *15*, **4544-4554**). The Bi_2S_3 layers are connected via **weaker intermolecular Bi...S bonds**, which resulting in the rapid growth of crystal along the *c*-axis, such as nanosheets or nanorods. The surface energies of the Bi_2S_3 crystal plane are 4.23×10^{-4} , 3.59×10^{-4} and 3.49×10^{-4} kJ m⁻² for the (100), (010) and (001) planes, respectively (*J. Phys. Chem. C*, **2009**, *113*, **36**). The result indicated (001) plane has the highest surface energy and fastest growth velocity. The big Bi_2S_3 particles preferred to grow into rods along [001] direction by consuming small particles during the ripening process. Bi_2S_3 nanorods grown along the *c* axis are rather stable. The intrinsic crystal structural feature determines the growth orientation of the Bi_2S_3 nanorods.

Comment 11: How were hydrodynamic sizes determined? "The diameter of $\text{Ti}_3\text{C}_2\text{T}_x/\text{Bi}_2\text{S}_3$ gradually decreased with the increase of $\text{Ti}_3\text{C}_2\text{T}_x$ content", Why? As

shown in TEM images, MXene nanosheets are larger than Bi₂S₃ nanorods, Also, as the method used for preparing MXene produces micron sized nanosheets, this statement seems to be incorrect.

Reply: Thank you very much for your professional suggestion. We are sorry for our negligence about the methods. Thus, we supplemented the description in revised manuscript in **Page 16**:

"The zeta potential and hydrodynamic sizes were measured by dynamic light scattering analysis (90 Plus/BI-MAS, Brookhaven, USA)."

For change of diameter for Bi₂S₃/Ti₃C₂T_{x-x}, the negative charged surface of Ti₃C₂T_x have strong electrostatic interaction with positively charge Bi³⁺. During synthesis process, the functionalized surface of Ti₃C₂T_x can reduce the nucleation barrier of bismuth sulfide and act as a nucleation site due to its capacity to bind bismuth ions. Thus, the diameter of Bi₂S₃/Ti₃C₂T_{x-x} is longer than that of Bi₂S₃. Then the released S²⁻ reacts with Bi³⁺ to form Bi₂S₃ crystal seeds. Bi₂S₃ nanocrystallites grow into nanorods. This heterogeneous of the Bi₂S₃ on the surface of Ti₃C₂T_x exist in the interface. Thus, the rate of nucleus formation and crystal growth are important factor for the microstructure (*Cryst Eng Comm*, 2012, 14, 3433–3440). As the increase of Ti₃C₂T_x solution, the rate of nucleation may increases. The rate of the crystal growth is lower than that of the nucleation, resulting in the decreased diameter.

For the size of MXene, we are sorry for our negligence about experimental detail. Before Bi(NO₃)₃·5H₂O was dissolved in Ti₃C₂T_x aqueous solution, Ti₃C₂T_x aqueous solution was sonicated by ultrasonic cell grinder. This process was under ice bath for 2 h at 20% amplitude in pulsed mode with 2s on 2s off, making the decreased size. Thus, we supplemented this information in **Page 15**:

"To synthesize the Bi₂S₃/Ti₃C₂T_x composites, Ti₃C₂T_x aqueous solution was sonicated by ultrasonic cell grinder. This process was under ice bath for 2 h at 20% amplitude in pulsed mode with 2s on 2s off."

Comment 12: As the MXene possesses O and F surface terminating groups, EDS mapping depicted in Figure 1 should also show these elements (Figure 1). The provided XPS verified the presence of O atoms.

Reply: Thanks a lot for your professional comment. Indeed, the surface of MXene possesses O and F terminating groups, thus EDS mapping should be provided. The corresponding EDS mapping have been supplemented in **Figure 1a** according to your advice. Moreover, we revised the description in the revised manuscript in **Page 4**: "The corresponding TEM element mapping analysis indicated that the C, Ti, O, F, Bi, and S elements were homogeneously distributed across the hybrid."

Fig. 1 | The characterizations of $\text{Bi}_2\text{S}_3/\text{Ti}_3\text{C}_2\text{T}_x$. a, TEM image of $\text{Bi}_2\text{S}_3/\text{Ti}_3\text{C}_2\text{T}_x$ and EDS elemental mappings.

Comment 13: Ti 2s peak in the XPS survey spectrum of the MB-5 sample is not visible, and the peak was not specified correctly.

Reply: Thank you very much for your professional suggestion. We are sorry for previous wide range of x axis, inducing the inaccurate mark. Following your advice, we have revised it in **Supplementary Fig. 9a**, these peaks have been marked in correct position.

Supplementary Fig. 9 | XPS measurements of as-synthesized samples. a, XPS survey scan.

Comment 14: Figure S8b: there should not be any O=C-O bonds in the structure of the MXene and MB-5.

Reply: Thank you very much for your professional suggestion. The samples of $\text{Bi}_2\text{S}_3/\text{Ti}_3\text{C}_2\text{T}_x$ -5 and $\text{Ti}_3\text{C}_2\text{T}_x$ contain O=C-O peak. The O=C-O bond is a common bond for different samples exposed to air, which was resulted from the contamination in the air. To better express this point, we have supplemented the manuscript in **Page S14 in Supplementary Informaiton:**

"The O=C-O peak may be from contamination for samples exposed to air^{8,9}."

In addition, the peak is very common in previous literatures.

8. Halim, J. et al. Transparent Conductive Two-Dimensional Titanium Carbide Epitaxial Thin Films. *Chem. Mater.* **26**, 2374-2381 (2014).
9. Li, Y. et al. A general Lewis acidic etching route for preparing MXenes with enhanced electrochemical performance in non-aqueous electrolyte. *Nat. Mater.* **19**, 894-899 (2020).

For examples:

- 1) *Nat. Mater.* **2020**, *19*, 894-899

Figure R3. XPS analysis of MS-Ti₃C₂T_x MXene. Spectra of C 1s.

2) *Adv. Mater.* 2018, 30, 1801629

Figure R4. XPS Survey spectrum of observed species C 1s.

Comment 15: Figure S19: It is hard to see the dead bacteria (red dots).

Reply: Thanks a lot for your professional comment. We are sorry for the unclear dead bacteria due to lower resolution. Thus, we have improved the resolution in our revised manuscript. The revised part is exhibited in **Page S29**.

Supplementary Fig. 22 | Bacterial live/dead fluorescence staining of *S. aureus*.

Bacterial live/dead fluorescence staining for $\text{Ti}_3\text{C}_2\text{T}_x$, Bi_2S_3 , $\text{Bi}_2\text{S}_3/\text{Ti}_3\text{C}_2\text{T}_x$ -2, $\text{Bi}_2\text{S}_3/\text{Ti}_3\text{C}_2\text{T}_x$ -5, $\text{Bi}_2\text{S}_3/\text{Ti}_3\text{C}_2\text{T}_x$ -7, and $\text{Bi}_2\text{S}_3/\text{Ti}_3\text{C}_2\text{T}_x$ -10 in vitro (green fluorescence represents live bacteria, red fluorescence represents dead bacteria). The scale bar is 100 μm .

Comment 16: The role of physical contact (10.1021/acsnano.6b00181 & 10.1021/acssuschemeng.8b03823) should be differentiated from the factor contributing to the antibacterial activity of the prepared nanocomposite.

Reply: Thank you very much for the professional suggestion. Indeed, the role of physical contact should be differentiated from the antibacterial performance. Thus, in our revised paper, our results demonstrated the antibacterial efficiency in the dark of $\text{Bi}_2\text{S}_3/\text{Ti}_3\text{C}_2\text{T}_x$ -x at the same condition by spread plate method (**Supplementary Fig. 21**), bacterial live/dead fluorescence staining (**Supplementary Fig. 22**), bacterial morphology (**Supplementary Fig. 23**). All of these data suggest that no significant antibacterial activity can be observed. So, the physical contact doesn't have any antibacterial effect in this system. In addition, the bacteria also were suspended in samples and cocultured under continuous shaking at 120 rpm and a constant temperature of 37 °C for 12 h to assess the antibacterial performance (**Supplementary Fig. 25**). There is also no obvious antibacterial activity. This also illustrated physical contact cannot play a key role in this system.

Supplementary Fig. 21 | Spread plate results and corresponding strain counts. a, b, Spread plate results (a) and corresponding strain counts (b) of *S. aureus*. **c, d,** Spread plate results (c) and corresponding strain counts (d) of *E. coli* treated with $Ti_3C_2T_x$, Bi_2S_3 , $Bi_2S_3/Ti_3C_2T_x-2$, $Bi_2S_3/Ti_3C_2T_x-5$, $Bi_2S_3/Ti_3C_2T_x-7$, and $Bi_2S_3/Ti_3C_2T_x-10$ under 808 nm irradiation for 10 min after 24 h co-culturing. Data are presented as mean \pm standard deviations from a representative experiment ($n = 3$ independent samples). Grey circles indicate the group of Ctrl, brown circles indicate the group of $Ti_3C_2T_x$, pink circles indicate the group of Bi_2S_3 , orange circles indicate the group of $Bi_2S_3/Ti_3C_2T_x-2$, green circles indicate the group of $Bi_2S_3/Ti_3C_2T_x-5$, blue circles indicate the group $Bi_2S_3/Ti_3C_2T_x-7$, purple circles indicate the group $Bi_2S_3/Ti_3C_2T_x-10$.

Supplementary Fig. 22 | Bacterial live/dead fluorescence staining of *S. aureus*. Bacterial live/dead fluorescence staining for $Ti_3C_2T_x$, Bi_2S_3 , $Bi_2S_3/Ti_3C_2T_x-2$, $Bi_2S_3/Ti_3C_2T_x-5$, $Bi_2S_3/Ti_3C_2T_x-7$, and $Bi_2S_3/Ti_3C_2T_x-10$ in vitro (green fluorescence represents live bacteria, red fluorescence represents dead bacteria). The scale bar is 100

μm .

Supplementary Fig. 23 | The SEM antibacterial images. a, b, SEM antibacterial images *S. aureus* (a) and *E. coli* (b) incubated with different samples of $\text{Ti}_3\text{C}_2\text{T}_x$, Bi_2S_3 , $\text{Bi}_2\text{S}_3/\text{Ti}_3\text{C}_2\text{T}_x-2$, $\text{Bi}_2\text{S}_3/\text{Ti}_3\text{C}_2\text{T}_x-5$, $\text{Bi}_2\text{S}_3/\text{Ti}_3\text{C}_2\text{T}_x-7$, and $\text{Bi}_2\text{S}_3/\text{Ti}_3\text{C}_2\text{T}_x-10$ in vitro. (The scale bar is $1 \mu\text{m}$).

Supplementary Fig. 25 | The antibacterial performance in the dark. Spread plate results and corresponding strain counts of *S. aureus* and *E. coli* for 12 h in the dark. Data are shown as mean \pm standard deviations; $n=3$ independent samples. Source data are provided as a Source Data file.

Comment 17: As antibacterial agents, many materials such as 2D nanomaterials of

black phosphorous, MoS₂, graphene oxide, carbon nitride, and Ag-based materials have been developed. These materials have exhibited excellent antibacterial properties. In this work, authors chose Ti₃C₂T_x MXene because of its photo-induced charge carriers. What are the advantage of Ti₃C₂T_x MXene compared with the mentioned materials?

Reply: Thank you very much for the professional suggestion. The comparison among 2D nanomaterials definitely helps further understand the role of Ti₃C₂T_x MXene. Among the 2D nanomaterials used in antibacterial agents, black phosphorous, metal dichalcogenides, graphene oxide, carbon nitride, and Ag-based materials are of special interest due to their structure, physical or chemical properties. Compared to these 2D materials, MXenes have extremely wide variety of chemical compositions and structural arrangement at the atomic scale. Ti₃C₂T_x MXene has been demonstrated to present some appealing characterization. First, it possesses excellent metallic conductivity ($> 10^4$ S cm⁻¹), which is beneficial to charge separation and transfer. Moreover, the surface functional terminations of Ti₃C₂T_x MXene can provide more active sites for semiconductor, improving the transmission kinetics of electrons. Its surface hydrophilicity and photothermal conversion property also broaden its application. Most important of all, the excellent biocompatibility can make sure the security in vivo.

Following the suggestion of the reviewer, we modified some statements in **Page 3**: "Ti₃C₂T_x MXene is an eco-friendly material with an excellent advantage due to its photo-induced charge carriers¹⁰⁻¹². The surface functional terminations of Ti₃C₂T_x MXene can provide more active sites for semiconductors. It is an attractive photoresponsive material, given its localized surface plasmon resonance (LSPR), strong absorption and conversion efficiencies for NIR light¹³, and high electrical conductivity, which can be utilized to regulate the energy bands of semiconductors, thereby enhancing the photocatalytic properties by accelerating the photo-induced charge transfer^{14, 15}."

In addition, we supplemented the statement in **Page 12**:

"Compared to other 2D nanomaterials of black phosphorous, MoS₂, graphene

oxide, carbon nitride, and Ag-based materials, MXene have excellent metallic conductivity, surface hydrophilicity and photothermal conversion property."

Comment 18: "No obvious diffraction peaks of $Ti_3C_2T_x$ were detected, which could be due to the low content and even distribution of $Ti_3C_2T_x$ ". This statement is questionable, as 10% of MXene is not a low content for collecting XRD spectra. Also, the distribution cannot affect the XRD patterns.

Reply: Thank you very much for the professional suggestion. We are sorry for our negligence. We carefully observed all the peaks of $Bi_2S_3/Ti_3C_2T_x$. The tiny peaks at about 61° for 2%, 5%, 7% and 10% $Bi_2S_3/Ti_3C_2T_x$ are ascribed to the (110) plane of $Ti_3C_2T_x$ MXenes. Thus, we revised the corresponding description in **Page S9 in Supplementary Information:**

"The XRD patterns of $Bi_2S_3/Ti_3C_2T_x$ with different content of $Ti_3C_2T_x$ (Supplementary Fig. 6b) showed that these peaks contain all the peaks of pristine Bi_2S_3 ^{4,5}. In addition, the tiny peaks at about 61° for 2%, 5%, 7% and 10% $Bi_2S_3/Ti_3C_2T_x$ are ascribed to the (110) plane of $Ti_3C_2T_x$ MXenes^{6, 7}"

6. Liao, Y. et al. 2D-layered Ti_3C_2 MXenes for promoted synthesis of NH_3 on P25 photocatalysts. *Appl. Catal., B* **273**, (2020).

7. Ghidui, M., Lukatskaya, M.R., Zhao, M.Q., Gogotsi, Y. & Barsoum, M.W. Conductive two-dimensional titanium carbide 'clay' with high volumetric capacitance. *Nature* **516**, 78-81 (2014).

Supplementary Fig. 6 | XRD patterns. **a**, XRD patterns of Ti_3AlC_2 and $\text{Ti}_3\text{C}_2\text{T}_x$. **b**, XRD patterns of Bi_2S_3 , $\text{Bi}_2\text{S}_3/\text{Ti}_3\text{C}_2\text{T}_x-2$, $\text{Bi}_2\text{S}_3/\text{Ti}_3\text{C}_2\text{T}_x-5$, $\text{Bi}_2\text{S}_3/\text{Ti}_3\text{C}_2\text{T}_x-7$, and $\text{Bi}_2\text{S}_3/\text{Ti}_3\text{C}_2\text{T}_x-10$.

Comment 19: Figure S4: "The clubbed structure of Bi_2S_3 almost completely covered the ultrathin $\text{Ti}_3\text{C}_2\text{T}_x$ nanosheets", I do not see any differences between the SEM images of Bi_2S_3 and MB-X.

Reply: Thank you very much for the professional suggestion. Indeed, we agree with the reviewer that no obvious differences can be observed for Bi_2S_3 and MB-x in SEM images. "The clubbed structure of Bi_2S_3 almost completely covered the ultrathin $\text{Ti}_3\text{C}_2\text{T}_x$ nanosheets." is only our speculation. We are sorry for our inaccurate statement. Thus, we revised this sentence in **Page S7 in Supplementary Information**.

"No obvious differences can be observed for Bi_2S_3 and MB-x in SEM images. It can be inferred that the clubbed structure of Bi_2S_3 may completely covered the ultrathin $\text{Ti}_3\text{C}_2\text{T}_x$ nanosheets."

Comment 20: To confirm the presence of the MXene in the structure of the nanocomposite, Raman spectroscopy is suggested. This technique is able to detect the MXene even with very low concentration of 0.05 wt% (see 10.1021/acsami.9b19960).

Reply: Thank you very much for the professional suggestion, which provide us a new method to confirm the presence of MXene. Raman spectroscopy was necessary to detect the presence of MXene. Following your advice, we have supplemented the corresponding data in **Supplementary Fig. 8**. And the descriptions have been shown in **Page S12 in Supplementary Information**:

"The Raman spectra of samples were provided in Supplementary Fig. 8, Raman bands located at 198 cm^{-1} , 393 cm^{-1} , and 615 cm^{-1} can be observed, matching with the Raman feature of $\text{Ti}_3\text{C}_2\text{T}_x$ ⁹. And the band at 965 cm^{-1} belongs to the Raman feature of Bi_2S_3 ¹⁰.

These Raman peaks of $\text{Ti}_3\text{C}_2\text{T}_x$ and Bi_2S_3 can be obviously observed in $\text{Bi}_2\text{S}_3/\text{Ti}_3\text{C}_2\text{T}_x$ -x, further verifying the presence of $\text{Ti}_3\text{C}_2\text{T}_x$ in the $\text{Bi}_2\text{S}_3/\text{Ti}_3\text{C}_2\text{T}_x$ -x."

9. Shamsabadi, A.A. et al. Pushing rubbery polymer membranes to be economic for CO_2 separation: embedment with $\text{Ti}_3\text{C}_2\text{T}_x$ MXene nanosheets. *ACS Appl Mater Interfaces* **12**, 3984-3992 (2020).
10. Xiao, Y., Cao, H., Liu, K., Zhang, S. & Chernow, V. The synthesis of superhydrophobic Bi_2S_3 complex nanostructures. *Nanotechnology* **21**, 145601 (2010).

Supplementary Fig. 8 | Raman spectra. Raman spectra of $\text{Ti}_3\text{C}_2\text{T}_x$, Bi_2S_3 , $\text{Bi}_2\text{S}_3/\text{Ti}_3\text{C}_2\text{T}_x$ -2, $\text{Bi}_2\text{S}_3/\text{Ti}_3\text{C}_2\text{T}_x$ -5, $\text{Bi}_2\text{S}_3/\text{Ti}_3\text{C}_2\text{T}_x$ -7, and $\text{Bi}_2\text{S}_3/\text{Ti}_3\text{C}_2\text{T}_x$ -10.

Comment 21: As the mass ratio of MXene is less than that of Bi_2S_3 , the name of the nanocomposite should be changed to $\text{Bi}_2\text{S}_3/\text{MXene}$.

Reply: Thank you very much for the professional suggestion. Indeed, the name of nanocomposite is not reasonable. Thus, we have revised the name of different samples in the revised manuscript. The obtained samples were labeled as $\text{Bi}_2\text{S}_3/\text{Ti}_3\text{C}_2\text{T}_x$ -2, $\text{Bi}_2\text{S}_3/\text{Ti}_3\text{C}_2\text{T}_x$ -5, $\text{Bi}_2\text{S}_3/\text{Ti}_3\text{C}_2\text{T}_x$ -7, and $\text{Bi}_2\text{S}_3/\text{Ti}_3\text{C}_2\text{T}_x$ -10, respectively.

Comment 22: In several places in this manuscript, " $\text{Ti}_2\text{C}_3\text{T}_x$ " should have been " $\text{Ti}_2\text{C}_3\text{T}_x$ ".

Reply: Thank you very much for the detail comment. We are very sorry for our negligence and we have carefully checked our manuscript. We have revised " $\text{Ti}_2\text{C}_3\text{T}_x$ " to " $\text{Ti}_2\text{C}_3\text{T}_x$ " in the revised manuscript.

Comment 23: The sentence "Based on the results, these results can be obtained as follows" needs to be rephrased.

Reply: Thank you very much for the detail comment. We are very sorry for our inaccurate expression and we have carefully revised our manuscript following your advice. We have already changed this sentence to "Based on the above experiments data, we can make the following conclusions." in the **Page 7**.

References

- [1] Rasool, Kashif, et al. "Antibacterial activity of $\text{Ti}_3\text{C}_2\text{T}_x$ MXene." *ACS nano* 10.3 (2016): 3674-3684.
- [2] Arabi Shamsabadi, Ahmad, et al. "Antimicrobial Mode-of-Action of Colloidal $\text{Ti}_3\text{C}_2\text{T}_x$ MXene Nanosheets." *ACS Sustainable Chemistry & Engineering* 6.12 (2018): 16586-16596.
- [3] Jastrzębska, A. M., et al. "In vitro studies on cytotoxicity of delaminated Ti_3C_2 MXene." *Journal of hazardous materials* 339 (2017): 1-8.
- [4] Lim, Gim Pao, et al. "Synthesis, characterization and antifungal property of $\text{Ti}_3\text{C}_2\text{T}_x$ MXene nanosheets." *Ceramics International* (2020).

- [5] Pandey, Ravi P., et al. "Ultra-high-flux and fouling-resistant membranes based on layered silver/MXene ($Ti_3C_2T_x$) nanosheets." *Journal of Materials Chemistry A* 6.8 (2018): 3522-3533.
- [6] Nie, Yan, et al. "MXene-Hybridized Silane Films for Metal Anticorrosion and Antibacterial Applications." *Applied Surface Science* (2020): 146915.
- [7] Mayerberger, Elisa A., et al. "Antibacterial properties of electrospun $Ti_3C_2T_z$ (MXene)/chitosan nanofibers." *RSC advances* 8.62 (2018): 35386-35394.
- [8] Jastrzębska, Agnieszka Maria, et al. "The atomic structure of Ti_2C and Ti_3C_2 MXenes is responsible for their antibacterial activity toward *E. coli* bacteria." *Journal of Materials Engineering and Performance* 28.3 (2019): 1272-1277.
- [9] Rozmysłowska-Wojciechowska, Anita, et al. "Engineering of 2D Ti_3C_2 MXene Surface Charge and Its Influence on Biological Properties." *Materials* 13.10 (2020): 2347.

Author's Response to Reviewer 2#

Original Comment: In this manuscript, the authors designed an interfacial Schottky junction of $\text{Ti}_3\text{C}_2\text{T}_x/\text{Bi}_2\text{S}_3$ with a built-in electric field resulting from the contact potential difference between $\text{Ti}_3\text{C}_2\text{T}_x$ and Bi_2S_3 . The self-driven charge transfer across the interface increases the local electron density on $\text{Ti}_3\text{C}_2\text{T}_x$ and boosts the charge transfer and separation. The $\text{Ti}_3\text{C}_2\text{T}_x/\text{Bi}_2\text{S}_3$ can kill 99.86% of *Staphylococcus aureus* and 99.92% of *Escherichia coli* with the assistance of hyperthermia within 10 min. The authors have done a lot of work in this manuscript, however, some problems still exist in this work and some formats need to be further clarified.

Reply: Thank you very much for your valuable and professional comments on our work. Thanks so much for offering us the opportunity to revise and improve our paper. We are sorry for our negligence to some formal error and we have carefully rechecked the manuscript to avoid these unnecessary wrong.

Comment 1: It is critically necessary to compare the bacterial-killing performance of previous reported photo-thermal systems with present $\text{Ti}_3\text{C}_2\text{T}_x/\text{Bi}_2\text{S}_3$ photocatalysts, the advantages of Bi_2S_3 over other semiconductors should be described. The biocompatibility of the $\text{Ti}_3\text{C}_2\text{T}_x/\text{Bi}_2\text{S}_3$ should be investigated.

Reply: Thank you very much for your professional suggestion. Yes, as you said, it is necessary to compare photo-thermal antibacterial system with our system. Thus, a systematic comparison was listed as follows. Usually, a separate photothermal antibacterial system needs to maintain higher temperature for long time (*10.1021/acsnano.0c05330*), which is used as the coating of mask and clothes. As for bacterial infection issue, such high temperature can damage normal tissues for a long time. Thus, the synergistic role of photothermal effect and other effect have become the hotspot of research in recent years. These effect contain nanozymes (*iScience 23, 101281; ACS Nano 2016, 10, 11000–11011*), ion release (*Nature Communication 10 (2019) 4490; Adv. Funct. Mater. 2019, 29, 1900143; Biomaterials 243 (2020) 119936*), drug release (*Nature communication 10 (2019) 4336*), physical damage (*Nano Lett.*

19 (2019) 5885–5896), and photodynamic therapy. Compared other antibacterial mechanism, photocatalysts involve a wide range, broadening antibacterial application.

Among these photocatalysts, many of these possess excellent biocompatibility, such as C_3N_4 , black phosphorus, TiO_2 , ZnO , MoS_2 , Bi_2S_3 , and so on. Compared with these 2D photocatalysts, Bi_2S_3 possesses high NIR absorption and narrow band energy. Moreover, it exhibited high heat conversion efficiency, resulting in excellent photothermal performance. In addition, we supplemented corresponding content in **Page 3** in Introduction part.

"Recently, many photocatalysts are used for photodynamic therapy, such as C_3N_4 , black phosphorus, TiO_2 , ZnO , MoS_2 , Bi_2S_3 , and so on. Compared with other semiconductors, Bi_2S_3 is considered as a potential photocatalytic material due to its n-type nature, direct narrow gap, and high absorption coefficient^{23, 24}."

Following your suggestion, we have supplemented the investigation of the $Ti_3C_2T_x/Bi_2S_{3x}$, including the cell viability and cell proliferation in vitro, and histological staining (heart, liver, spleen, lung, and kidneys) in vivo, which can be seen in **Supplementary Fig. 28** and **Supplementary Fig. 33** in **Supplementary Information**.

$Bi_2S_3/Ti_3C_2T_x$ exhibited excellent cell viability and did not affect cell proliferation. Moreover, the excellent biosafety in vivo can be observed.

Supplementary Fig. 28 | The cell biocompatibility *in vitro* of materials. a, Cell viability for 1 and 3 d in the dark. **b,** Cell viability for 1 and 3 d with 808 nm for 10 min. **c,** Cell fluorescence staining of control, $Ti_3C_2T_x$, Bi_2S_3 , $Bi_2S_3/Ti_3C_2T_x-2$, $Bi_2S_3/Ti_3C_2T_x-5$, $Bi_2S_3/Ti_3C_2T_x-7$ and $Bi_2S_3/Ti_3C_2T_x-10$. The scale bars are 50 μ m. Data are presented as mean \pm standard deviations from a representative experiment (n = 3 independent samples). Grey circles indicate the group of Ctrl, brown circles indicate the group of $Ti_3C_2T_x$, pink circles indicate the group of Bi_2S_3 , orange circles indicate the group of $Bi_2S_3/Ti_3C_2T_x-2$, green circles indicate the group of $Bi_2S_3/Ti_3C_2T_x-5$, blue circles indicate the group $Bi_2S_3/Ti_3C_2T_x-7$, purple circles indicate the group $Bi_2S_3/Ti_3C_2T_x-10$.

Supplementary Fig. 33 | H&E staining of heart, liver, spleen, lung, and kidney treated with control, 3M wound dressing, $Ti_3C_2T_x$, Bi_2S_3 , and $Bi_2S_3/Ti_3C_2T_x-5$. The scale bars are 200 μ m.

Comment 2: The size of Bi_2S_3 was about 200 nm, does the size of Bi_2S_3 affect the performance?

Reply: Thank you very much for your professional suggestion. This is a very good question. The length of Bi_2S_3 rods may have an effect on the separation efficiency of photoexcited electron-hole pairs. According to your comments, we prepared the different length of $Bi_2S_3/Ti_3C_2T_x-5$, the rate of electron generation was measured. The result demonstrated the photogenerated electrons gradually increased with the increase

of the length of Bi_2S_3 rods. This may be attributed to the different incident photon-to-current conversion efficiency. The different length of TiO_2 nanowires also exhibited the similar results (*ACS Nano*, 2012, 6, 5060-5069).

In Supplementary Information in our revised paper, we supplemented the statement in **Page S40**:

"To demonstrate the effect of Bi_2S_3 rod lengths on the rate of photogenerated electrons, the different lengths of Bi_2S_3 was prepared. As shown in **Supplementary Fig. 30a**, the different lengths can be observed from particles to long rods (more than 1 μm). PL spectra can reflect the separation efficiency of photogenerated electrons and holes. The longer rods showed a lower photogenerated electron-hole recombination rate, indicating the efficient transfer and separation of photoexcited electron-hole. Similarly, there was an increasing tendency in photocurrent with the increase of length. PL spectra and photocurrent density showed that the length of Bi_2S_3 had an influence on the separation of electron-hole pairs. This similar result was also demonstrated by TiO_2 nanowires length²⁶."

26. Hwang, Y. J., Hahn, C., Liu, B., & Yang, P. Photoelectrochemical properties of TiO_2 nanowire arrays: a study of the dependence on length and atomic layer deposition coating. *ACS Nano* 6, 5060-5069, (2012).

Supplementary Fig. 30 | The characterization and photocatalytic properties of

Bi₂S₃/Ti₃C₂T_x-5 with different rods. a, SEM images **b**, PL spectra tested with 325 nm excitation wavelength from 350 nm to 600 nm. **c**, Photocurrent response under 808 nm NIR irradiation. Source data are provided as a Source Data file.

Comment 3: The authors stated that the synergistic impacts of the ROS and photothermal effect killed the bacteria rapidly and effectively. The author needs to study which free radicals play a role in this system. Which is more important, ROS or photothermal effect?

Reply: Thank you very much for your professional suggestion. This is a very good question. Thus, we supplemented a series of data to illustrate this question.

Firstly, the types of free radicals, including $\cdot\text{OH}$, $^1\text{O}_2$, and $\cdot\text{O}_2^-$, was detected by electron spin resonance (ESR) spectra, 1, 3-Diphenylisobenzofuran (DPBF), and Nitro Blue Tetrazolium (NBT) method, respectively. The results indicate that Bi₂S₃/Ti₃C₂T_x can produce $\cdot\text{OH}$ and $\cdot\text{O}_2^-$ under light irradiation. No obvious $^1\text{O}_2$ can be detected.

Thus, in our revised paper, we supplemented the corresponding description in **Page 7**: "Under 808 nm NIR light irradiation, the electron spin resonance (ESR) spectra, Nitro Blue Tetrazolium (NBT) method, and 1,3-Diphenylisobenzofuran (DPBF) (Fig. 3b, 3c, and Supplementary Fig. 15) were used to measure $\cdot\text{OH}$, $\cdot\text{O}_2^-$, $^1\text{O}_2$, respectively. As it can be seen in Fig. 3b and 3c, obviously, large amounts of $\cdot\text{OH}$ and $\cdot\text{O}_2^-$ were detected under 808 nm NIR irradiation, while no signals for these two species were detected in the dark, indicating that $\cdot\text{OH}$ and $\cdot\text{O}_2^-$ can be generated under 808 nm NIR light^{38, 39}. In addition, NBT can react with $\cdot\text{O}_2^-$ to produce monoformazan (MF) that exhibits a maximum absorbance at 530 nm. As shown in Supplementary Fig. 14a, compared with the dark group, Bi₂S₃/Ti₃C₂T_x-5 under 808 nm NIR light irradiation had high absorption, suggesting the production of $\cdot\text{O}_2^-$. DPBF result indicated that almost nothing $^1\text{O}_2$ was detected during 10 min NIR light irradiation (Supplementary Fig. 15b)."

In **Page 17**, we added corresponding description:

"The superoxide radicals ($\cdot\text{O}_2^-$) were measured by Nitro Blue Tetrazolium (NBT)

after 808 NIR light irradiation. The amount of $\cdot\text{O}_2^-$ can be examined by the absorption spectra of monoformazan (MF) because MF is produced through the reaction between $\cdot\text{O}_2^-$ and NBT dissolved in dimethyl sulfoxide (DMSO) (2.5×10^{-5} mol/L) solution. 1, 3-diphenylisobenzofuran (DPBF) was used to detect the production of the $^1\text{O}_2$ during light irradiation. The absorption of DPBF at 416 nm was decreased when reacting with $^1\text{O}_2$."

Fig. 3 | The mechanism of interfacial engineering based on work function. b, ESR spectra of hydroxyl radicals of $\text{Bi}_2\text{S}_3/\text{Ti}_3\text{C}_2\text{T}_x-5$. **c,** ESR spectra of superoxide radical of $\text{Bi}_2\text{S}_3/\text{Ti}_3\text{C}_2\text{T}_x-5$.

Supplementary Fig. 15 | $\cdot\text{O}_2^-$ and $^1\text{O}_2$ characterization. a, The absorption spectra of MF in the dark or 808 nm light. **b,** The detection of $^1\text{O}_2$ using DPBF as detector under light irradiation.

To verify the role of ROS and photothermal effect to antibacterial performance, the antibacterial rate of ROS alone or photothermal effect alone was assessed by spread

plate results. As shown in **Supplementary Fig. 24**, under ROS effect, the antibacterial rate against *S. aureus* and *E. coli* were 37.06%, 43.47%, respectively. The antibacterial efficiency of 48.51% against *S. aureus* and 54.88% against *E. coli* can be observed under the action of single photothermal effect. The sum of the individual photothermal and photodynamic antibacterial effects (85.57% for *S. aureus* and 98.35% for *E. coli*) is less than the synergistic antibacterial effect (99.86% for *S. aureus* and 99.92% for *E. coli*). Thus, the synergistic effect of ROS and photothermal effect can reach excellent antibacterial performance. In our revised paper, we have modified the manuscript as following:

In **Page 11**, we have supplemented "The antibacterial efficacy of single ROS or photothermal effect alone demonstrated that the synergy of ROS and photothermal effect achieved the best antibacterial performance (Supplementary Fig. 24)."

In **Page 19**, the corresponding experimental methods have been supplemented: "The antibacterial activity of ROS alone was assessed in a constant temperature water bath at 25 °C. The antibacterial activity of photothermal effect alone was measured by the addition of ascorbic acid."

In **Page S32 in Supplementary Information**, we have added the statement:

"To verify the synergistic effect of ROS and photothermal effect, the antibacterial performance was assessed. Under single ROS effect, the antibacterial rate against *S. aureus* and *E. coli* was 37.06% and 43.47%, respectively. In addition, under photothermal effect alone, the antibacterial efficiency of 48.51% against *S. aureus* and 54.88% against *E. coli* can be observed. The sum of the individual photothermal and photodynamic antibacterial effects (85.57% for *S. aureus* and 98.35% for *E. coli*) is less than the synergistic antibacterial effect (99.86% for *S. aureus* and 99.92% for *E. coli*). Thus, the synergy of ROS and photothermal effect can achieve much better antibacterial performance than single ROS or photothermal effect alone."

Supplementary Fig. 24 | The antibacterial performance for ROS effect alone or thermal effect alone. Spread plate results and corresponding strain counts of *S. aureus* and *E. coli*. Data are shown as mean \pm standard deviations; $n = 3$ independent samples.

Comment 4: The antibacterial properties of the $\text{Ti}_3\text{C}_2\text{T}_x/\text{Bi}_2\text{S}_3$ was investigated under the irradiation of the 808 nm? What about its performance under the irradiation with other wavelength?

Reply: Thank you very much for your professional suggestion. According to your advice, we investigate the antibacterial performance under 660 nm light and simulated

solar irradiation. $\text{Bi}_2\text{S}_3/\text{Ti}_3\text{C}_2\text{T}_x$ exhibited inferior antibacterial performance under these conditions. The antibacterial efficacy of the prepared material was only 65.08% and 77.17% against *S. aureus* and *E. coli*, respectively, under 660 nm light irradiation (**Supplementary Fig. 26**). Under simulated solar irradiation, the antibacterial rate was only 45.95% and 58.34% against *S. aureus* and *E. coli*, respectively (**Supplementary Fig. 27**). The corresponding data were supplemented as following:

In **Page 11**: "The antibacterial performance of materials under 660 nm light or simulated solar irradiation was also assessed. As shown in Supplementary Fig. 26 and Supplementary Fig. 27, $\text{Bi}_2\text{S}_3/\text{Ti}_3\text{C}_2\text{T}_x-5$ exhibited inferior antibacterial performance under these conditions than under 808 nm light irradiation."

Additionally, the experimental methods have also been supplemented in **Page 19**: "The antibacterial performance of materials under 660 nm or simulated solar irradiation was also examined by spread plate method."

In **Supplementary Information, Page S35**: "To verify the antibacterial performance of the samples under 660 nm light irradiation, photothermal curves and the yield of ROS were measured. As shown in Supplementary Fig. 26, the temperature only reached to 55.2 °C within 10 min and the yield of ROS was much lower than that of $\text{Bi}_2\text{S}_3/\text{Ti}_3\text{C}_2\text{T}_x-5$ under 808 nm light irradiation. The antibacterial efficacy only reached to 65.08% and 77.17% against *S. aureus* and *E. coli*, which is lower than that under 808 nm light irradiation. This is attributed to the low temperature and ROS yield under 660 nm light."

In **Supplementary Information, Page S37**: "The antibacterial activity of $\text{Bi}_2\text{S}_3/\text{Ti}_3\text{C}_2\text{T}_x-5$ under simulated solar irradiation was also assessed. As shown in **Supplementary Fig. 27**, $\text{Bi}_2\text{S}_3/\text{Ti}_3\text{C}_2\text{T}_x-5$ exhibited poor antibacterial performance. The antibacterial efficacy was only 45.95% and 58.34% against *S. aureus* and *E. coli*, respectively, which is much lower than that under 808 nm light irradiation."

Supplementary Fig. 26 | The antibacterial performance of Bi₂S₃/Ti₃C₂T_x-5 under 660 nm (0.7 W cm⁻¹). **a**, Photothermal curves under 10 min light. **b**, ROS production with DCFH fluorescence probe (200 ppm). **c**, Spread plate results and corresponding strain counts of *S. aureus* and *E. coli*. Data are presented as mean ± standard deviations from a representative experiment.

Supplementary Fig. 27 | The antibacterial performance of Bi₂S₃/Ti₃C₂T_x-5 under solar irradiation (0.2 W cm⁻²). Spread plate results and corresponding strain counts of *S. aureus* and *E. coli*. Data are presented as mean ± standard deviations from a representative experiment.

Comment 5: In-situ XPS should be done to evidence the built-in electric field.

Reply: Thank you very much for your professional suggestion. According to your advice, *in-situ* XPS under 808 nm light irradiation was measured. As shown in **Supplementary Fig. 9**, it can be observed that, the binding energy of Ti-C for Bi₂S₃/Ti₃C₂T_x-5 under 808 nm light shifted negatively with reference to that of ex situ spectra. Conversely, the peaks of Bi 4f and S 2p shifted positively. The shifts demonstrated that the photogenerated electrons in Bi₂S₃ can transfer to Ti₃C₂T_x under 808 nm light irradiation.

Following your comments, in our revised paper, we have supplemented the statements in **Page S14 in Supplementary Information**: "In addition, the charge transfer under 808 nm light irradiation was evidenced by the in situ XPS. The binding energy of Ti-C for Bi₂S₃/Ti₃C₂T_x-5 under 808 nm light irradiation shifted negatively with reference to that in ex situ spectra. Conversely, the peaks of Bi 4f and S 2p shifted positively. This shift demonstrates that the photogenerated electrons in Bi₂S₃ can transfer to Ti₃C₂T_x

under 808 nm light irradiation."

In this manuscript, the existence of built-in electric field is due to the formation of Schottky junction between Bi_2S_3 and $\text{Ti}_3\text{C}_2\text{T}_x$. The formation of built-in electric field does not require any external conditions. The difference in Fermi level between metal $\text{Ti}_3\text{C}_2\text{T}_x$ and semiconductive Bi_2S_3 can result in the charge flow until the system reaches to equilibrium. Experimental evidences (**Fig. 3a**) and density functional theory calculations (**Fig. 4a and 4b**) have demonstrated that the work function of $\text{Ti}_3\text{C}_2\text{T}_x$ is larger than that of $\text{Ti}_3\text{C}_2\text{T}_x$, that is to say, the Fermi energy of the $\text{Ti}_3\text{C}_2\text{T}_x$ surface was lower than that of $\text{Ti}_3\text{C}_2\text{T}_x$, which meets the conditions for Schottky's contact. Once Bi_2S_3 contacts with $\text{Ti}_3\text{C}_2\text{T}_x$, the electrons will transfer from Bi_2S_3 to $\text{Ti}_3\text{C}_2\text{T}_x$. When the equilibrium reaches, the electronic density of the side of $\text{Ti}_3\text{C}_2\text{T}_x$ will increase and the electronic density of the side of Bi_2S_3 will decrease. The local difference of electronic density can cause built-in electric field at the interface of Bi_2S_3 and $\text{Ti}_3\text{C}_2\text{T}_x$. In addition, the peak shift of ex situ XPS also verified this point (**Supplementary Fig. 9**).

Supplementary Fig. 9 | In situ and ex situ XPS measurements of as-synthesized samples. b-c, High-resolution of C 1s (b), Bi 4f and S 2p (c).

Fig. 3 | The mechanism of interfacial engineering based on work function. **a**, UPS spectra measured by He I ($h\nu = 21.22$ eV) spectra the secondary electron cutoff of $\text{Ti}_3\text{C}_2\text{T}_x$ and valence band of Bi_2S_3 and MB-5 Schottky catalyst with respect to the Fermi level. **c**, Energy scheme before and after contact of $\text{Ti}_3\text{C}_2\text{T}_x$ and n-type Bi_2S_3 .

Fig. 4 | DFT calculation of the $\text{Bi}_2\text{S}_3/\text{Ti}_3\text{C}_2\text{F}_2$ and $\text{Bi}_2\text{S}_3/\text{Ti}_3\text{C}_2\text{O}_2$ structure. **a**, **b**, Electrostatic potential along z axis of $\text{Bi}_2\text{S}_3/\text{Ti}_3\text{C}_2\text{F}_2$ (**a**) and $\text{Bi}_2\text{S}_3/\text{Ti}_3\text{C}_2\text{O}_2$ (**b**) interface. The red dashed lines denote E_{vac} , the gray line represent E_{F} . **c**, **d**, The differential charge density of $\text{Bi}_2\text{S}_3/\text{Ti}_3\text{C}_2\text{F}_2$ (**c**) and $\text{Bi}_2\text{S}_3/\text{Ti}_3\text{C}_2\text{O}_2$ (**d**) interface. Pink and yellow colors denote electron excess and deficiency area, respectively.

Author's Response to Reviewer 3#

Original Comment: The manuscript entitled "Interfacial engineering of $Ti_3C_2T_x$ MXene/ Bi_2S_3 based on work function for rapid photo-excited bacteria-killing" (NCOMMS-20-22666-T) designed an interfacial Schottky junction of $Ti_3C_2T_x/Bi_2S_3$ with a built-in electric field. The prepared $Ti_3C_2T_x/Bi_2S_3$ exhibited an excellent antibacterial performance towards *Staphylococcus aureus* and *Escherichia coli* under 808 nm near-infrared light (NIR) radiation. The manuscript is well structured and the mechanisms of antibacterial performance are provided in detail. This study is of great importance and it promotes the development of the eradication of bacterial infection. Overall, the experiments have been conducted adequately. I believe that the paper is worth to be published in Nature Communications after revision based on the following:

Reply: We express our sincere thanks to the reviewer for your positive comments and valuable suggestions, which will definitely help us further improve the quality of this work. "I believe that the paper is worth to be published in Nature Communications after revision".

Comment 1: - Page 5, "The interface space only was 2.34 Å, confirming close contact." Please add related references to support this expression.

Reply: Thanks a lot for the reviewer's detail suggestion. The theoretical calculation result from density function theory showed the average interface spacing between Bi_2S_3 and $Ti_3C_2T_x$ was 2.34 Å. We have added two references in **Page 5**. "A stable crystal structure was formed and the average interface space was about 2.34 Å, confirming close contact^{32, 33}."

32. Wang, X.-D. et al. In situ construction of a Cs_2SnI_6 perovskite nanocrystal/ SnS_2 nanosheet heterojunction with boosted interfacial charge transfer. *J. Am. Chem. Soc.* **141**, 13434-13441 (2019).

33. Yang, Y. et al. In situ grown single - atom Cobalt on polymeric Carbon Nitride with bidentate ligand for efficient photocatalytic degradation of refractory antibiotics. *Small* **16**, 2001634 (2020).

Comment 2: Page 10, the author wrote that "Moreover, the homogeneous formation of Bi_2S_3 on the surface of $\text{Ti}_3\text{C}_2\text{T}_x$ is beneficial for rapid and uniform heating." How to understand and verify the homogeneous formation?

Reply: Thanks for the reviewer's professional question. We are sorry for our inaccurate statement and we have revised the corresponding descriptions in our revised manuscript following your advice, which have been shown in **Page 10**:

"Moreover, the homogeneous **distribution** of Bi_2S_3 on the surface of $\text{Ti}_3\text{C}_2\text{T}_x$ is beneficial for rapid and uniform heating."

The homogeneous distribution can be proved by EDS mapping data in Figure 1a. It can be seen that the element of Ti, C, O, F, Bi, and S uniformly distributed across the $\text{Bi}_2\text{S}_3/\text{Ti}_3\text{C}_2\text{T}_x$. The result verified the homogeneous formation of Bi_2S_3 on the surface of $\text{Ti}_3\text{C}_2\text{T}_x$. The description was shown in **Page 4**: "The corresponding TEM element mapping analysis indicated that the C, Ti, O, F, Bi, and S elements were homogeneously distributed across the hybrid." The corresponding result was exhibited in **Figure 1a**.

Figure 1a. TEM image of $\text{Bi}_2\text{S}_3/\text{Ti}_3\text{C}_2\text{T}_x$ and EDS elemental mappings.

Comment 3: How about the stability of the prepared $\text{Ti}_3\text{C}_2\text{T}_x/\text{Bi}_2\text{S}_3$?

Reply: Thank you very much for the professional suggestion. Indeed, we agree with the reviewer that the stability of the material plays a vital role. Thus, the stability of prepared $\text{Bi}_2\text{S}_3/\text{Ti}_3\text{C}_2\text{T}_x$ was characterized by XPS result. The comparison of

$\text{Bi}_2\text{S}_3/\text{Ti}_3\text{C}_2\text{T}_x$ before and after one month can explain that $\text{Bi}_2\text{S}_3/\text{Ti}_3\text{C}_2\text{T}_x$ exhibited good stability.

In Page S15 in Supplementary Information, we have added Supplementary Fig. 10 and supplemented the statement: "In order to systematically evaluate the stability of $\text{Bi}_2\text{S}_3/\text{Ti}_3\text{C}_2\text{T}_x$, XPS was measured before and after one month. There was no change for the signal of Ti, C, Bi, and S from XPS survey scan (Supplementary Fig. 10a). In addition, both of them exhibited C-C, C-O, Ti-C and O=C-O bonds from the high-resolution spectra of C 1s (Supplementary Fig. 10b). Similar, the high-resolution spectra of Bi 4f and S 2p reveal four peaks assigned to Bi 4f_{7/2}, S 2p_{3/2}, S 2p_{1/2} and Bi 4f_{5/2} (Supplementary Fig. 10c). These evidence indicated that $\text{Bi}_2\text{S}_3/\text{Ti}_3\text{C}_2\text{T}_x$ possess good stability."

Supplementary Fig. 10 | XPS measurements of $\text{Bi}_2\text{S}_3/\text{Ti}_3\text{C}_2\text{T}_x$ -5 before and after a month. a, XPS survey scan. b, c, High-resolution of C 1s (b) and Bi 4f and S 2p (c).

Comment 4: Recent important reviews and researches can be added to better summarize the background (Introduction). Related references are listed below:

- [1] Chem. Eng. J. 2019, 370, 1087-1100.
- [2] Coordin. Chem. Rev. 385 (2019) 44-80.
- [3] Chem. Eng. J. 2020, 382, 122840.
- [4] Appl. Catal. B Environ. 2020, 267, 118651.

Reply: Thanks a lot for your professional comment. We would like to thank the referee for this valuable comment, which offers us a better understanding of the background. These recent important reviews and researches have been supplemented in Introduction part of our revised paper. The paper (Coordin. Chem. Rev. 385 (2019) 44-80) was

inserted as reference [8]. The paper (*Appl. Catal. B Environ.* 2020, 267, 118651) was cited as reference [9]. These references were cited in **Page 3**. Besides, the paper (*Chem. Eng. J.* 2019, 370, 1087-1100) were cited as reference [26]. The paper (*Chem. Eng. J.* 2020, 382, 122840) were cited as reference [27]. In addition, the relevant reference (*Appl. Catal. B Environ.* 2019, 258) also has been cited as reference [14]. These references were briefly introduced in **Page 3**:

"However, practically, the efficacies of ROS and heat generation drastically decrease due to poor electron mobility and rapid recombination of electrons and holes in single-component semiconductors^{8,9}."

"It is an attractive photoresponsive material, given its localized surface plasmon resonance (LSPR), strong absorption and conversion efficiencies for NIR light¹³, and high electrical conductivity, which can be utilized to regulate the energy bands of semiconductors, thereby enhancing the photocatalytic properties by accelerating the photo-induced charge transfer^{14,15}."

"However, the narrow band gap easily gives rise to rapid recombination of the photogenerated electrons and holes, tremendously reducing the utilization of the former²⁵⁻²⁷."

8. Huang, D. et al. Artificial Z-scheme photocatalytic system: What have been done and where to go? *Coordin. Chem. Rev.* **385**, 44-80 (2019).
9. Huang, D. et al. Zn_xCd_{1-x}S based materials for photocatalytic hydrogen evolution, pollutants degradation and carbon dioxide reduction. *Appl. Catal. B* **267**, 118651 (2020).
14. Yang, Y. et al. Ti₃C₂ Mxene/porous g-C₃N₄ interfacial Schottky junction for boosting spatial charge separation in photocatalytic H₂O₂ production. *Appl. Catal. B* **258**, (2019).
26. Chen, S. et al. Modifying delafossite silver ferrite with polyaniline: Visible-light-response Z-scheme heterojunction with charge transfer driven by internal electric field. *Chem. Eng. J.* **370**, 1087-1100 (2019).
27. Chen, S. et al. In-situ synthesis of facet-dependent BiVO₄/Ag₃PO₄/PANI photocatalyst with enhanced visible-light-induced photocatalytic degradation

performance: Synergism of interfacial coupling and hole-transfer. *Chem. Eng. J.* **382**, 122840 (2020).

Lastly, we would like to thank the Editor and all the Reviewers again for their time and efforts in helping us improve the quality of this manuscript. We hope that our responses are satisfactory and the revised manuscript could meet the standard of *Nature Communications*.

REVIEWERS' COMMENTS

Reviewer #1 (Remarks to the Author):

My comments were addressed adequately in the revision; apart from the three very minor comments listed below, I am satisfied by the responses and the changes that were made. In my view, the authors carried out a thorough study of a novel material.

1. The following new sentence on p.11 needs a rearrangement of words: "In summary, none of the as-prepared materials for ten minutes under static conditions showed antibacterial properties."
2. "The XRD patterns of Bi₂S₃/Ti₃C₂T_x with different content of Ti₃C₂T_x (Supplementary Fig. 6b) showed that these peaks contain all the peaks of pristine Bi₂S₃,5. ...", "different content" "different contents".
3. On p.7, "Based on the above experiments data, we can make the following conclusions.", "experiments data"  "wet-lab results".

Reviewer #2 (Remarks to the Author):

The author answered the comments of the reviewers very well, and added relevant experiments and data, so I agree to publish.

Response to Reviewer 1#

Original Comment: My comments were addressed adequately in the revision; apart from the three very minor comments listed below, I am satisfied by the responses and the changes that were made. In my view, the authors carried out a thorough study of a novel material.

Reply: Thank you for your positive evaluation. We express our sincere thanks to the reviewer for the positive comments and valuable suggestions, which have definitely improved the quality of this work. We will carefully modify our manuscript according your comments. Thank you.

1. The following new sentence on p.11 needs a rearrangement of words: "In summary, none of the as-prepared materials for ten minutes under static conditions showed antibacterial properties."

Reply: Thank you for your professional suggestion. According to your advice, we have modified the sentence in Page 12 in the revised paper: "In summary, none of the as-prepared materials showed antibacterial properties for ten minutes under static conditions."

2. "The XRD patterns of $\text{Bi}_2\text{S}_3/\text{Ti}_3\text{C}_2\text{T}_x$ with different content of $\text{Ti}_3\text{C}_2\text{T}_x$ (Supplementary Fig. 6b) showed that these peaks contain all the peaks of pristine $\text{Bi}_2\text{S}_3^{4,5}$", "different content" "different contents".

Reply: Thank you for your professional suggestion. We are sorry for our negligence. We have modified this sentence in Page S9: "The XRD patterns of $\text{Bi}_2\text{S}_3/\text{Ti}_3\text{C}_2\text{T}_x$ with different contents of $\text{Ti}_3\text{C}_2\text{T}_x$ (Supplementary Fig. 6b) showed that these peaks contain all the peaks of pristine $\text{Bi}_2\text{S}_3^{4,5}$."

3. On p.7, "Based on the above experiments data, we can make the following conclusions.", "experiments data"  "wet-lab results".

Reply: Thank you for your professional suggestion. We are sorry for our inaccurate

expression. We have modified this sentence in Page 7: "Based on the above wet-lab results, we can make the following conclusions."

Response to Reviewer 2#

Original Comment: The author answered the comments of the reviewers very well, and added relevant experiments and data, so I agree to publish.

Reply: Thank you very much for your valuable and professional comments on our work. Thanks so much for the positive evaluation.